# Superior Colliculus to VTA pathway controls orienting response and influences social interaction in mice

Clément Solié [1,2,3], Alessandro Contestabile [1,3], Pedro Espinosa [1], Stefano Musardo [1], Sebastiano Bariselli[1], Chieko Huber [1], Alan Carleton [1] & Camilla Bellone [1✉]

Social behaviours characterize cooperative, mutualistic, aggressive or parental interactions that occur among conspecifics. Although the Ventral Tegmental Area (VTA) has been identified as a key substrate for social behaviours, the input and output pathways dedicated to specific aspects of conspecific interaction remain understudied. Here, in male mice, we investigated the activity and function of two distinct VTA inputs from superior colliculus (SC-VTA) and medial prefrontal cortex (mPFC-VTA). We observed that SC-VTA neurons display social interaction anticipatory calcium activity, which correlates with orienting responses towards an unfamiliar conspecific. In contrast, mPFC-VTA neuron population activity increases after initiation of the social contact. While protracted phasic stimulation of SC-VTA pathway promotes head/body movements and decreases social interaction, inhibition of this pathway increases social interaction. Here, we found that SC afferents mainly target a subpopulation of dorsolateral striatum (DLS)-projecting VTA dopamine (DA) neurons (VTA$^{DA}$-DLS). While, VTA$^{DA}$-DLS pathway stimulation decreases social interaction, VTA$^{DA}$-Nucleus Accumbens stimulation promotes it. Altogether, these data support a model by which at least two largely anatomically distinct VTA sub-circuits oppositely control distinct aspects of social behaviour.

[1] Department of Basic Neuroscience, University of Geneva, 1 Rue Michel-Servet, 1205 Genève, Switzerland. [2] Present address: Brain Plasticity Unit, CNRS UMR 8249, ESPCI, PSL Research University, Paris, France. [3] These authors contributed equally: Clément Solié, Alessandro Contestabile. ✉email: Camilla.bellone@unige.ch

The term "social interaction" describes interactions with conspecifics, which are highly complex behaviors guided by both environmental and internal states. When exploring novel social settings, individuals constantly integrate external cues and internal states to orient towards conspecifics. These processes prepare the subject to deal with the decision to approach or avoid the conspecific[1]. At a biological level, several brain regions are functionally relevant for conspecific interaction[1]. Therefore, to fully understand the complexity of this behavior, the neurocircuitry underlying specific aspects of social interaction needs to be precisely investigated.

Because of its role in reward and motivation, the ventral tegmental area (VTA) and dopamine (DA) have long been implicated in conspecific interaction[2,3]. Indeed, DA release peaks upon initial contact with conspecifics and habituates upon subsequent presentation of the same stimulus[3]. More recently, it has been shown that the activity of DA neurons encodes key aspects of social interaction[4]. In fact, while stimulation of VTA DA neurons projecting to the NAc (VTA$^{DA}$-NAc) increases interaction[4], chemogenetic inhibition of VTA DA neurons attenuates both the exploration and the reinforcing properties of novel conspecifics in mice[5]. However, the circuit mechanisms that modulate DA neuron activity during interactions with novel conspecifics remain largely unknown.

According to a theory developed by Sokolov, novel stimuli generate orienting responses that depend on the significance of the stimulus and habituate upon stimulus repetition[6]. Importantly, Pavlov stressed that "orienting responses are necessary for terms of organism's survival. If it were absent, the life of the animal would be in constant danger". Thus, orienting responses can be described as complex reactions to significant events and form the basis of natural selective attention[7]. In this framework, social stimuli can be considered as environmental salient stimuli that capture attention and generate orienting responses towards a conspecific. Subsequently, the individual makes predictions about the valence of an eventual interaction and adapts its behavior to either initiate or avoid social contact. Thus, orienting responses are one of the first and necessary steps to engage in social behavior. However, the neural circuits that control orienting responses towards conspecific and their role in social behavior remain unknown.

Input-output mapping of the circuit architecture of VTA revealed diverse sources of excitatory inputs[8]. Superior Colliculus (SC) is an interesting input to VTA in the context of social behavior, because it is an evolutionarily conserved midbrain structure for sensory information processing and motor functions. Furthermore, SC is not only activated by biological salient stimuli independently of their valence[9] but also controls orientation and spatial attention[10–14]. Interestingly it has been previously shown that intermediate and deep layers of SC form synaptic contacts with DA and non-DA neurons of the substantia nigra pars compacta (SNc) providing the short-latency visual inputs[15,16]. A different population of SC neurons provides input to VTA[17,18]. While SC excitatory projections to VTA GABA neurons control defensive behavior, SC inputs to VTA DA neurons might contribute to reinforcement learning[19]. However, whether SC has a pivotal role in conspecific interaction by conveying information to VTA neurons during social orienting remains unknown.

Different subpopulations of midbrain DA neurons regulate emotional, cognitive and motor functions by targeting the striatum and the cortex. It is generally assumed that VTA DA neurons project to the ventral part of the striatum, while DA neurons in the SN send projections to the dorsal striatum. However, a gradient distribution of DA cells projecting to the ventromedial, central and dorsolateral striatum has been observed[17]. Through a spiral input-output connection between midbrain and striatum as well as afferents from limbic regions to VTA neurons[8], information flows from limbic to motor circuits providing a mechanism by which motivation can influence decision-making and action performance[20]. Thus, this crosstalk between circuits would be important to adjust behavior in the function of novel information. Whether different loops within the midbrain circuit play a different role in conspecific interaction is still largely unknown.

Here, we show that the neuronal activity of identified SC neurons projecting to VTA (SC-VTA) increases during orienting response toward unfamiliar conspecifics. Optogenetic stimulation and inhibition of SC-VTA pathway disturbs orienting response towards social stimuli and interfere with conspecific interaction. Interestingly, the activity of the medial Prefrontal cortex (mPFC) to VTA input pathway (mPFC-VTA) increases during conspecific interaction but not during orienting response toward the social stimulus. Remarkably, we found that DA neurons receiving SC projections are located in the lateral VTA and mainly provide inputs to the dorsolateral striatum (DLS). Finally, optogenetic stimulation of the VTA$^{DA}$-DLS pathway affects conspecific interaction similarly to SC-VTA stimulation.

## Results

**VTA receives projection from SC**. To characterize anatomical projections from SC to VTA, we injected an adeno-associated virus (AAV) expressing a yellow fluorescent protein (eYFP; AAV5-hSyn-eYFP) in the SC (Fig. 1a, b) and observed SC axons in the VTA (Fig. 1b). To identify the anatomical position of SC neurons projecting to the VTA, we injected a retrograde virus (ssAAV-retro/2-hSyn1-chl-tdTomato; Fig. 1c, d) in the VTA. This virus allowed us to label the soma of VTA-projecting neurons and revealed that they are predominantly located in the intermediate and deep SC layers (dSC, Fig. 1e). Furthermore, immunohistochemistry analysis of retrogradely labelled tdTomato neurons, indicated that 88% of SC neurons projecting to the VTA are CaMKII positive (Fig. 1f, g), suggesting a prevalent excitatory connection between SC and VTA.

To explore the functional connectivity of SC-VTA projections, we injected an AAV5-hSyn-ChR2-eYFP in SC and performed whole-cell patch-clamp recordings. First, we confirmed that the SC neurons followed the light stimulation protocol (Fig. 1h). In the second set of experiments, we utilized transgenic mice expressing the Cre-recombinase under the control of the promoter of either DA transporter (DAT-Cre) or glutamic acid decarboxylase (GAD65-Cre). These mice received VTA infusions of AAV5-Ef1α-DIO-mCherry to fluorescently label either dopaminergic (VTA-DAT$^+$) or GABAergic (VTA-GAD$^+$) neurons, and a second SC infusion with AAV5-hSyn-ChR2-eYFP (Fig. 1i). Optogenetic stimulation of axons in the VTA evoked monosynaptic excitatory postsynaptic currents (EPSCs) (Fig. 1j, k) and revealed connectivity of SC with both VTA-DAT$^+$ (25.7%) and GAD$^+$ (42.3%) neurons (Fig. 1l). Remarkably, the percentage of evoked EPSC was higher than IPSC for VTA-DAT$^+$ cells (25.7% vs 4.4%), further strengthening the notion of a prevalent excitatory connection between SC and VTA (Fig. 1m). Interestingly, we observed that SC-stimulation responsive DAT$^+$ and GAD$^+$ neurons were located in the lateral and medial portion of the VTA (Fig. 1n), respectively. Taken together, these results show that distinct neuronal populations within the VTA receive inputs from SC.

**SC-VTA neurons are activated during the orientation test**. SC to VTA pathway has been previously associated with innate defensive behavior[16] and to acute dark induction of

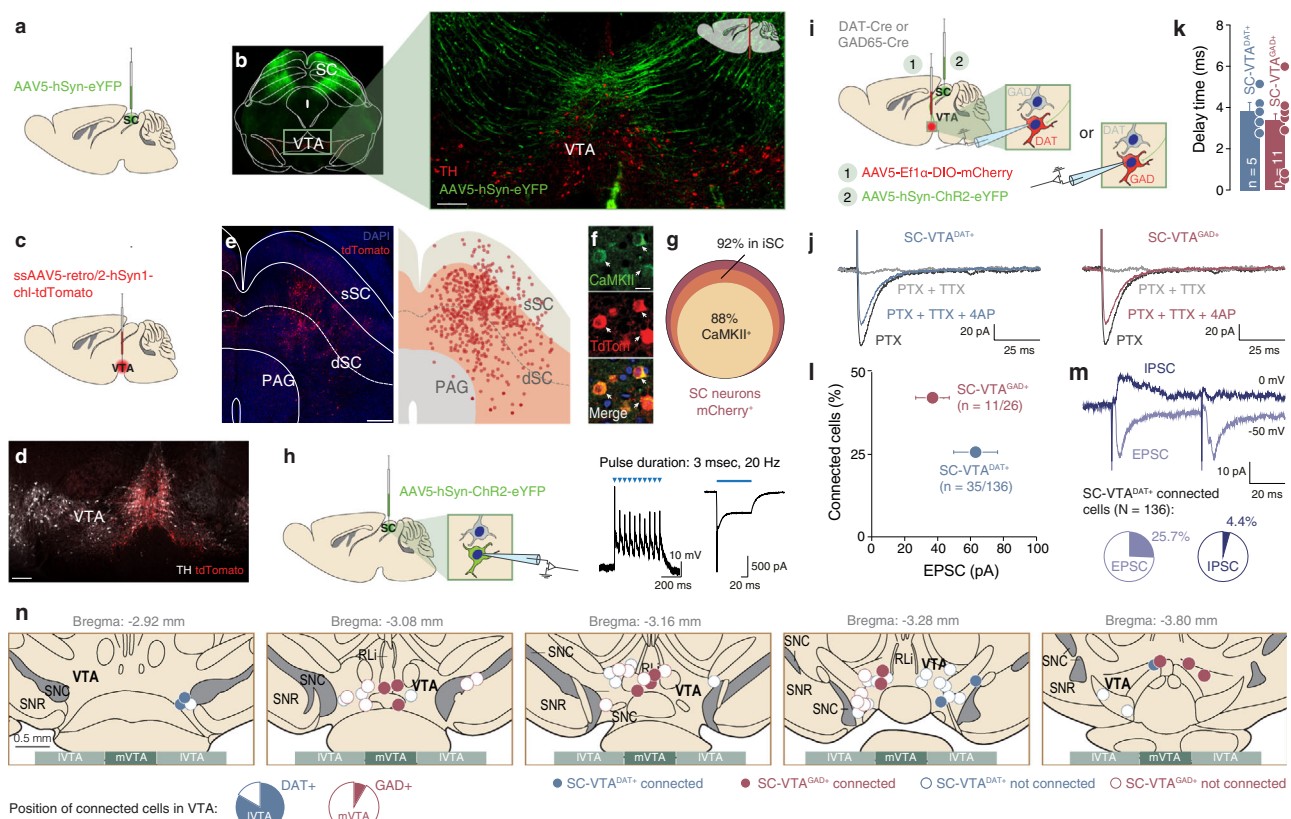

**Fig. 1 Anatomo-functional connectivity from SC to VTA dopaminergic and GABAergic neurons. a** Schema of injection in the Superior Colliculus (SC) with AAV5-hSyn-eYFP. **b** Right: Representative coronal image of immunostaining experiments against Tyrosine Hydroxylase (TH) enzyme (in red) performed on midbrain slices of adult mice infected with AAV5-hSyn-eYFP (green) in the SC. The projecting fibers from SC to VTA are visible. Left: Image at higher magnification of the coronal slice (scale bar: 100 μm). The fibers project from SC to VTA (this experiment was reproduced at least three times). **c** Schema of injection in the VTA with ssAAV5-retro/2-hSyn1-chl-tdTomato. **d** Representative coronal image of ssAAV5-retro/2-hSyn1-chl-tdTomato injection site in the VTA. Immunostaining labelled TH-positive neurons (scale bar: 200 μm. This experiment was reproduced at least three times). **e** Left: Representative image of the infected cells with ssAAV5-retro/2-hSyn1-chl-tdTomato in the SC (scale bar: 200 μm. This experiment was reproduced at least three times). Right: schema reporting the position of tdTomato positive cells in the superficial (sSC) and intermediate/deeper layers (dSC) of the SC and in the periaqueductal grey (PAG) for six infected brains. **f** Representative image of immunostaining against $Ca^{2+}$/calmodulin-dependent protein kinase II (CAMKII) in the SC and infected cells with ssAAV5-retro/2-hSyn1-chl-tdTomato (scale bar: 10 μm. This experiment was reproduced at least three times). **g** Quantification of tdTomato infected cells in the SC for six infected brains. The cells are preferentially located in the intermediate layer of the SC and are predominantly CAMKII+. **h** Whole-cell patch-clamp from SC ChR2-expressing neurons. Protocol of stimulation indicates that SC neurons follow 20 Hz light stimulation protocol. **i** Schema of injection in the SC with AAV5-hSyn-ChR2-eYFP and patch of the VTA DAT+ and GAD+ neurons (depending on the mice line). **j** Example traces of optogenetically elicited excitatory postsynaptic currents (EPSCs) in VTA DAT+ and GAD+ neurons in presence of PTX, TTX, and 4AP. **k** Delay time of the EPSCs for VTA DAT+ and GAD+ neurons. **l** Quantification of connected cells from SC onto VTA DAT+ and GAD+ neurons in relation of the amplitude of EPSCs. **m** Upper panel: Example traces of optogenetically elicited EPSCs or inhibitory postsynaptic currents (IPSCs) in VTA DAT+ neurons. Lower panel: Percentage of evoked EPSCs or IPSCs in the VTA DAT+ neurons connected with the SC. **n** Position of some patched VTA DAT+ and GAD+ neurons. SC-VTA^DAT+ connected neurons are mainly in the lateral part of the VTA (lVTA) while SC-VTA^GAD+ connected neurons are more medially located (mVTA). n indicates number of cells. All the data are shown as the mean ± s.e.m. as error bars. Source data are provided as a Source Data file.

wakefulness[21], suggesting an important role of this pathway in rapid adaptations to environmental changes. Since dSC layer neurons control orienting responses and approach behavior[22], we investigated whether SC to VTA pathway may be implicated in orienting response toward salient moving stimuli. We targeted the SC neurons projecting to the VTA by injecting a retrograde AAV encoding Cre virus (AAVrg-Ef1α-mCherry-IRES-Cre) in the VTA and a Cre-dependent AAV encoding GCaMP6s in the SC (AAV9-hSyn-FLEX-GCamp6s-WPRE-SV40; Fig. 2a). We then implanted an optic fiber in the SC (Fig. 2b) to record $Ca^{2+}$ transients from VTA-projecting neurons during social orienting. In order to dissect orienting responses in mice, we developed a test where the experimental mouse is positioned in the center of a circular arena where it can turn the head and rotate the body

(Fig. 2c). After habituation, we placed a sex-matched juvenile conspecific in the external circular corridor (Fig. 2c). During the experiment, the nose point of the experimental mouse and the body-center point of both mice were tracked and the head orientation angle (ω) towards the conspecific was determined (Fig. 2c). We identified three events during the test in which the experimental mouse was oriented towards the stimulus (ω = 0°): the ipsilateral (1) or contralateral (2) head/body turn in direction of the conspecific and (3) the stimulus passive crossing (Fig. 2d, e and Supplementary Fig. 1a, b). Peri-event time histogram (PETH) revealed a significant increase in normalized ΔF/F (Z-score) during ipsi-recorded orienting response (Fig. 2e). Interestingly, the increase in $Ca^{2+}$ signals was transient and returned to baseline immediately after orientation even if the alignment lasted

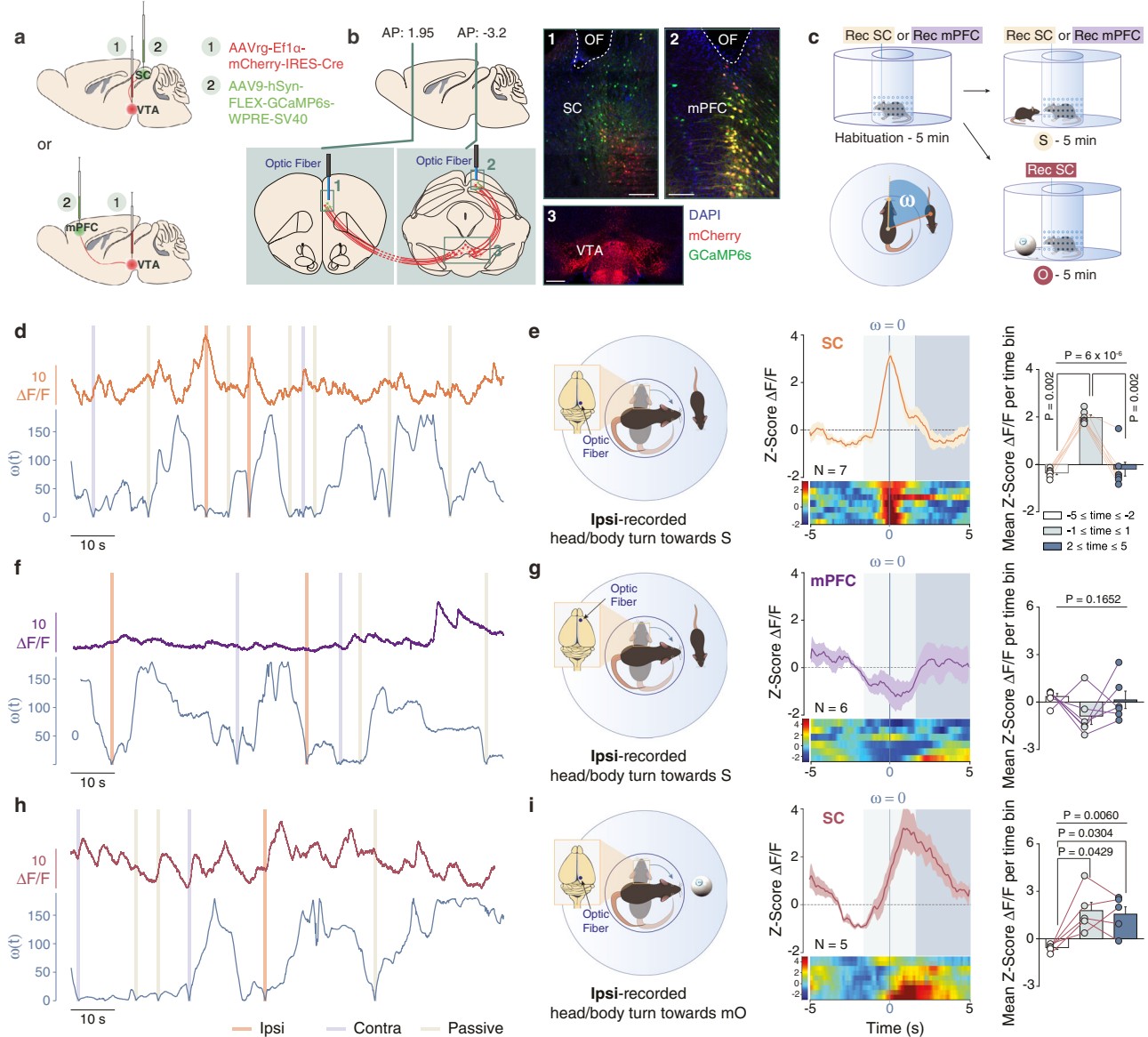

**Fig. 2 Calcium activity of SC to VTA-projecting neurons during social and non-social orientation tests. a** Schema of injections of AAVrg-Ef1α-mCherry-IRES-Cre in the VTA and AAV9-hSyn-FLEX-GCaMP6s-WPRE-SV40 in the SC or mPFC. Representative coronal images of midbrain slices of adult mice infected with AAVrg-Ef1α-mCherry-IRES-Cre in the VTA (**b** panel 3, scale bar: 100 μm) and AAV9-hSyn-FLEX-GCaMP6s-WPRE-SV40 in the SC (**b** panel 1, scale bar: 50 μm) and in the mPFC (**b** panel 2, scale bar: 50 μm). The location of the optic fiber (OF) is indicated. Similar viral expression and OF location were observed in all the mice that performed the experiment described in Figs. 2c, 4b. **c** Schema of the social and non-social orientation test. Bottom left panel: Schema representing the points and vectors used for the calculation of the oriented angle towards the stimulus (ω). **d, f, h** Example traces. ΔF/F signals recorded in SC- or mPFC-VTA-projecting neurons aligned with instantaneous head orientation ω(t) during orientation test. Ipsi-recorded and contra-recorded head/body turn episodes are reported as well as the passive crossing events. **e** Left panel: Schema of the ipsi-recorded head/body turns towards the social stimulus (S). Middle panel: Peri-event time histogram (PETH) of normalized ΔF/F for SC-VTA-projecting neurons, centered on ipsi-recorded orientation towards social stimulus. Right panel: Mean ΔF/F (Z-score) before, during and after ipsi-recorded orientation. Repeated measures (RM) one-way ANOVA (Events main effect: $F_{(2,6)} = 44.34$, $P < 0.0001$) followed by Bonferroni-Holm post-hoc test correction. **g** Left panel: Schema of the ipsi-recorded head/body turns towards the social stimulus (S). Middle panel: PETH of normalized ΔF/F for mPFC-VTA-projecting neurons, centered on ipsi-recorded orientation towards social stimulus. Right panel: Mean ΔF/F (Z-score) before, during and after ipsi-recorded orientation. RM one-way ANOVA (Events main effect: $F_{(2,5)} = 2.17$, $P = 0.1652$). **i** Left panel: Schema of the ipsi-recorded head/body turns towards the moving object (mO). Middle panel: PETH of normalized ΔF/F for SC-VTA-projecting neurons, centered on ipsi-recorded orientation towards moving object. Right panel: Mean ΔF/F (Z-score) before, during and after ipsi-recorded orientation. RM one-way ANOVA (Events main effect: $F_{(2,4)} = 10.34$, $P = 0.0060$) followed by Bonferroni-Holm post-hoc test correction. N indicates the number of mice. All the data are shown as the mean ± s.e.m. as error bars or error bands. Source data are provided as a Source Data file.

more than one second (Fig. 2e). No significant increase in normalized ΔF/F was observed during contra-recorded orienting response or passive crossing events (Supplementary Fig. 1c, d). Moreover, when we isolated the rearing events (Supplementary Fig. 1e), we did not observe any significant variation in $Ca^{2+}$ signal (Supplementary Fig. 1f) indicating that the SC-VTA pathway was active during movement directed toward a moving stimulus.

VTA supports a vast variety of animal behavior by combining information from many different brain regions. To investigate the specificity of information coming from the SC to the VTA, we used the same experimental protocol to record $Ca^{2+}$ transients from medial prefrontal cortex (mPFC) neurons projecting to the VTA (Fig. 2a–c). In order to compare the SC-VTA and mPFC-VTA pathways, a cohort of C57BL/6 J was randomly split, and surgeries and behavioral experiments were performed at the same time for both groups of mice. Interestingly, no significant increase in normalized ΔF/F was observed during ipsi- or contra-recorded orientation responses, passive crossing or rearing events (Fig. 2f, g and Supplementary Fig. 1g–i).

Finally, to verify whether the activation of SC-VTA pathway was generalizable to other salient moving stimuli, we replaced the juvenile conspecific with a moving ball which was programmed to move similarly to a conspecific (Fig. 2c). Remarkably, ipsi-recorded orienting response towards the moving object also induced an increase in $Ca^{2+}$ transients (Fig. 2h–i).

These data indicate that the SC-VTA pathway encodes the orienting response towards salient stimuli, and strongly suggest that time-lock activity during head-turning must occur in order to generate a correct orienting response.

**Optogenetic manipulation of the SC-VTA neurons disrupts orienting response towards social stimuli.** To prove the role of SC-VTA pathway in the orientation test, we injected AAV5-hSyn-eYFP (control), blue-light-sensitive ChR2 (AAV-hSyn-ChR2-eYFP) or red light-sensitive optogenetic inhibitor Jaws (AAV-hSyn-Jaws-GFP or AAV-hSyn-eYFP as control) in the SC. In a second time, we implanted an optic fiber over the VTA to stimulate or inhibit SC axon terminals (Fig. 3a–c, examples of optic fibers' tips localization after post-hoc validation are reported in Supplementary Fig. 2a). As a control, we tested that Jaws activation induces terminal inhibition by recording light-induced EPSCs in the VTA (Supplementary Fig. 4a–c). We analyzed the position of the social stimulus relative to the experimental mouse and we measured the time spent by the conspecific within the experimental subject in the frontal field of view (head-oriented angle < 45° per side; Fig. 3d). We observed that during the first minute of the light ON condition, SC-VTA ChR2-expressing mice spent less time oriented towards the social stimulus (Fig. 3e). Moreover, analysis of the time spent by the social stimulus in the frontal field revealed habituation for the eYFP-expressing mice group in the light OFF and ON condition (Fig. 3f). On the other hand, the stimulus initially spent less time in the frontal field of ChR2-expressing mice compared to controls and habituation was not observed (Fig. 3f). Contrary to the control condition, Jaws-expressing mice during the light ON epoch did not habituate and 76.9% of them increased the orientation time towards the social stimulus during the second minute (Fig. 3e, f). Interestingly, we observed that immediately after optogenetic stimulation, ChR2-expressing mice stopped the action that they were performing and increased head/body turns (Fig. 3g, h).

When we substitute the juvenile conspecific with a moving ball (Supplementary Fig. 2b), eYFP- and ChR2-expressing animals did not habituate to the stimulus between the 1st and the 2nd min (Supplementary Fig. 2c). Even though the difference between the

time pass in the frontal field during ON and OFF epoch is not significantly different, we observed that ChR2-expressing mice tend to pass less time oriented towards the moving ball during the first minute of the light ON epoch (Supplementary Fig. 2c).

These data demonstrate that stimulation of SC-VTA pathway increases head/body turns and decreases the time passed by the experimental mouse oriented towards a moving stimulus.

**SC-VTA pathway is activated during orienting responses in the course of social interaction.** To probe the activity of SC to VTA pathway during direct social interaction, we targeted the SC neurons projecting to the VTA by injecting a retrograde AAV encoding Cre virus (AAVrg-Ef1α-mCherry-IRES-Cre) in the VTA and a Cre-dependent AAV encoding GCaMP6s in the SC (AAV9-hSyn-FLEX-GCamp6s-WPRE-SV40; Fig. 4a). We then implanted an optic fiber in the SC, and recorded ΔF/F during free social interaction with a juvenile sex-matched conspecific (Fig. 4b, c). We observed a transient increase in SC-VTA pathway $Ca^{2+}$ activity starting before nose-to-nose (Fig. 4d) and nose-to-body contact (Fig. 4e). Similarly, to the pattern observed during the orientation test, the $Ca^{2+}$ signals returned to baseline immediately after the interaction (Fig. 4c–e). On the contrary, no changes were observed before or after passive contacts initiated by the conspecific (Fig. 4f). Remarkably, we noticed that not all nose-to-nose or nose-to-body events were followed by an increase of $Ca^{2+}$ intake and that peaks of $Ca^{2+}$ transients were also observed when the two animals were distant from each other (examples in Fig. 4c). We, therefore, decided to perform a more precise analysis that would take into consideration both the head orientation angle (ω) and the distance between the mice (Fig. 4c; Proximal: when the distance between the nose of the experimental mouse and the body center of the stimulus is less than 5 cm, distal: when this distance is more than 5 cm). Importantly, and similarly to the results obtained in the orientation test, we revealed that both proximal and distal ipsi-recorded orienting responses towards the conspecific were aligned with an increase of activity in the SC-VTA pathway (Fig. 4g, h). On the other hand, no significant increase in normalized ΔF/F was observed during passive crossing events (Fig. 4i) or proximal and distal contra-recorded orienting responses (Fig. 4j, k).

Finally, in order to investigate the specificity of the information coming from the SC to the VTA, we performed a social interaction test while recording the calcium activity in the mPFC-VTA pathway (Supplementary Fig. 3a, b). Interestingly, the activity of the mPFC-VTA pathway shows different temporal dynamics compared to the SC-VTA pathway. Indeed, normalized ΔF/F increased after nose-to-nose and nose-to-body contacts and persisted during interaction (Supplementary Fig. 3c–e).

These data suggest that the SC-VTA pathway is recruited during the head/body turn to control the orienting response toward the conspecific while the mPFC-VTA pathway is activated during the interaction.

**SC-VTA pathway bidirectionally controls social interaction.** Social behavior is the result of a sequence of different features: stop of previous actions, orientation toward the stimulus, interaction, and end of the contacts. Since we show that the SC to VTA pathway is involved in orientation toward salient stimuli, we decided to test the functional relevance of this pathway during social interaction. Consequently, we injected blue-light-sensitive ChR2 (AAV-hSyn-ChR2-eYFP or AAV-hSyn-eYFP as control) in the SC. In a second surgery, we implanted an optic fiber over the VTA to stimulate SC axon terminals (Fig. 5a–c). After recovery, we placed the mice in a homecage-like arena and, after 3 min of baseline, we exposed the experimental mouse to an unfamiliar

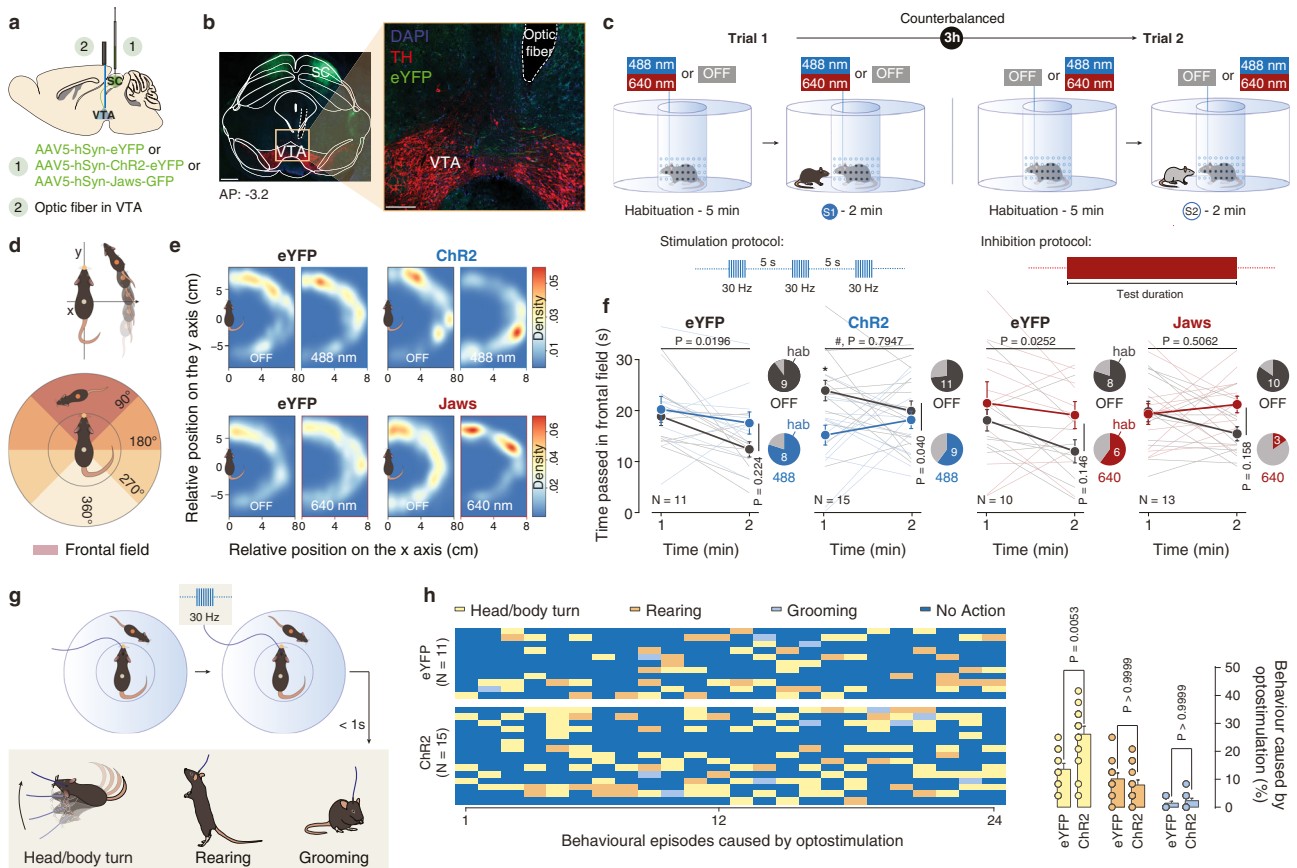

**Fig. 3 Optogenetic manipulation of SC-VTA pathway alters orienting response. a** Schema of injections sites in SC with AAV5-hSyn-eYFP, AAV5-hSyn-ChR2-eYFP, or AAV5-hSyn-Jaws-GFP, and optic fiber implantation above the VTA. **b** Representative image of coronal midbrain slices of adult mice infected with AAV5-hSyn-eYFP (green) in the SC. In red is visible the immunostaining anti-Tyrosine Hydroxylase (TH) (Left panel: scale bar = 500 μm. Right panel: image at higher magnification, scale bar = 100 μm). Similar viral expression and OF location were observed in all the mice that performed the experiment described in Figs. 3c, 5b. **c** Top panel: Schema of the social orientation test. The eYFP-, ChR2- and Jaws-expressing mice oriented towards two different unfamiliar mice under both stimulation conditions. Bottom left panel: Stimulation and inhibition protocols. 8 pulses of 488 nm light (30 Hz) were separated by 5 s in the light ON condition. Bottom right panel: Continuous inhibition was instead provoked with 640 nm light. **d** Upper panel: schema representing the relative position of the social stimulus when the center body point of the experimental animals is fixed at (0, 0) and the nose point is fixed along the y-axis. Lower panel: schema representing the position of the frontal field. **e** Heatmaps reporting the relative position of the social stimulus during orientation test for the 1st and 2nd minute in the different conditions. **f** Time passed with the social stimulus in the frontal field for the 1st and 2nd minute of the social orienting test in light and no-light conditions. RM two-way ANOVA (eYFP$_{488 nm}$: Light main effect: $F_{(1,10)} = 1.683$, $P = 0.2236$; Time main effect: $F_{(1,10)} = 7.711$, $P = 0.0196$; Light × Time Interaction: $F_{(1,10)} = 1.254$, $P = 0.2890$. ChR2: Light main effect: $F_{(1,14)} = 5.138$, $P = 0.0398$; Time main effect: $F_{(1,14)} = 0.070$, $P = 0.7947$; Light × Time Interaction: $F_{(1,14)} = 4.868$, $P = 0.0446$. eYFP$_{640 nm}$: Light main effect: $F_{(1,9)} = 2.537$, $P = 0.1456$; Time main effect: $F_{(1,9)} = 7.185$, $P = 0.0252$; Light × Time Interaction: $F_{(1,9)} = 0.6144$, $P = 0.4533$. Jaws: Light main effect: $F_{(1,12)} = 2.266$, $P = 0.1581$; Time main effect: $F_{(1,12)} = 0.4696$, $P = 0.5062$; Light × Time Interaction: $F_{(1,12)} = 4.572$, $P = 0.0538$) followed by Bonferroni's multiple comparisons post-hoc test. Pie charts represent the percentage of mice that decrease the orientation between 1st and 2nd minute. **g** Possible behaviors caused by the optostimulation and observed less than 1 sec after a burst of light. **h** Left panel: identification and report of the behavioral episodes caused by optostimulation. Right panel: quantification of the behavioral episodes caused by optostimulation. Two-way ANOVA (Behavior main effect: $F_{(1.874,43.09)} = 60.95$, $P < 0.0001$; Group main effect: $F_{(1, 23)} = 3.684$, $P = 0.0674$; Behavior × Group Interaction: $F_{(2, 46)} = 11.29$, $P = 0.0001$) followed by Bonferroni's multiple comparisons post-hoc test. N indicates the number of mice. # indicates significantly different interaction. All the data are shown as the mean ± s.e.m. as error bars. Source data are provided as a Source Data file.

conspecific for 2 min (Fig. 5b). The experiment was then repeated with another unfamiliar conspecific after 3 h delay with the opposite light condition (counterbalanced, Fig. 5b). Interestingly, the photostimulation of SC-VTA pathway in ChR2-expressing mice decreased the overall time interaction without affecting the eYFP-control group (Fig. 5d). The time-course of the interaction with the unfamiliar conspecific revealed that eYFP-expressing mice quickly habituated to the stimulus and spent less time investigating the conspecific through the second minute (Fig. 5e). Habituation was reflected also in the time that the social stimulus spent in the frontal field (Fig. 5e). Interestingly, ChR2-expressing mice decreased their interaction time and the time spent with the

social stimulus in the frontal field during the first minute of the light ON epoch (Fig. 5e). As observed in the orientation test, habituation did not occur for ChR2-expressing mice (ON condition, Fig. 5e). Analysis of the behavior caused by a single burst of optogenetic stimulation showed an increase in ChR2-induced head/body turn compared to controls while rearing, grooming and retreat behavior were not affected (Fig. 5f). Remarkably, optogenetic stimulation also caused a significant increase in the interruptions of social interaction performed by ChR2-expressing mice (Fig. 5f). These data show that bursting activation of the SC-VTA pathway increases the number of head body turns and subsequently decreases social interaction and orienting behaviors.

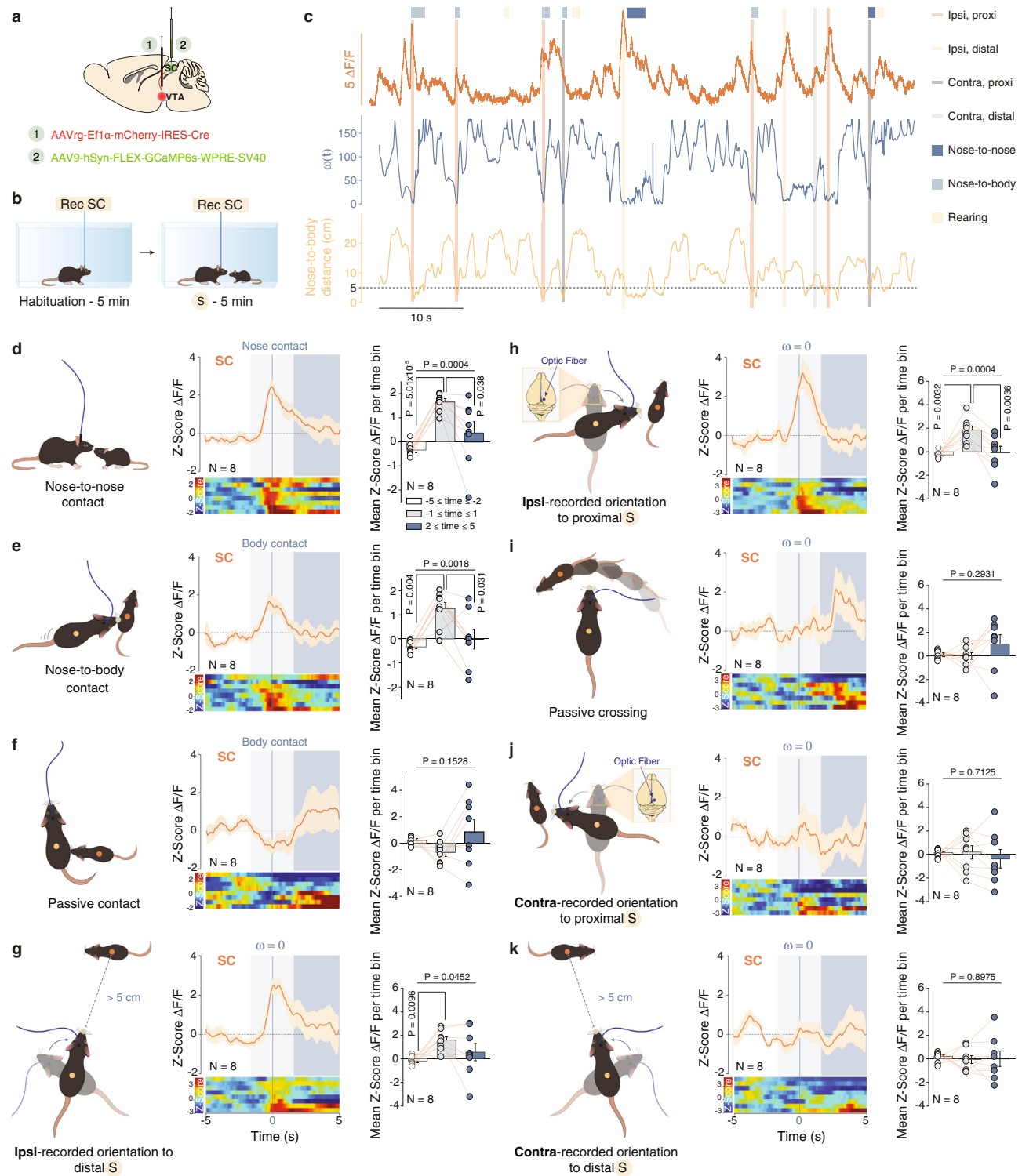

To test the effect of SC to VTA pathway inhibition during social interaction, we injected the red light-sensitive optogenetic inhibitor Jaws (AAV-hSyn-Jaws-GFP or AAV-hSyn-eYFP as control) in the SC in a new batch of mice (Supplementary Fig. 4d). Contrary to photostimulation, photoinhibition of SC terminal in the VTA significantly increased investigation time in Jaws-expressing mice without affecting the eYFP-control group (Supplementary Fig. 4e). Importantly, Jaws-expressing mice increased their interaction time during the first minute of the light ON epoch (Supplementary Fig. 4f) and no-light-dependent difference was observed in the time spent with the social stimulus

in the frontal field (Supplementary Fig. 4f), possibly due to the continuous and constant light inhibition.

To better dissect the behavioral consequences of SC-VTA pathway manipulation, we analyzed detailed aspects of social and non-social behavior sequences during both light OFF and light ON epochs (Supplementary Fig. 4g). Although we did not find differences in rearing behavior, photostimulation induced less following and nose-to-nose interaction (Supplementary Fig. 4h–j), while photoinhibition elicited opposite changes in following behavior between light ON and OFF conditions (Supplementary Fig. 4k–m). These data suggest that SC-VTA pathway bidirectionally controls free social interaction.

**Fig. 4 Calcium activity of SC and mPFC to VTA-projecting neurons during free social interaction. a** Schema of injections of AAVrg-Ef1α-mCherry-IRES-Cre in the VTA and AAV9-hSyn-FLEX-GCaMP6s-WPRE-SV40 in the SC. **b** Schema of free social interaction test. **c** Example traces. ΔF/F signals recorded in SC-VTA-projecting neurons aligned with instantaneous head orientation ω(t) and the distance between the nose of the experimental mouse and the gravity center of the stimulus mouse during the free social interaction test. Ipsi-recorded (proximal and distal) and contra-recorded (proximal and distal) head/body turn episodes are reported as well as the manual scoring of the nose-to-nose, nose-to-body and passive contacts. **d** Left: Schema of nose-to-nose contact. Middle: Peri-event time histogram (PETH) of normalized ΔF/F for SC-VTA-projecting neurons, centered on nose-to-nose contacts. Right: Mean ΔF/F (Z-score) before, during and after nose-to-nose. RM one-way ANOVA (Events main effect: $F_{(2,7)} = 14.15$, $P = 0.0004$) followed by Bonferroni-Holm post-hoc test correction. **e** Left: Schema of nose-to-body contact. Middle: PETH of normalized ΔF/F for SC-VTA-projecting neurons, centered on nose-to-body contacts. Right: Mean ΔF/F (Z-score) before, during and after nose-to-body. RM one-way ANOVA (Events main effect: $F_{(2,7)} = 10.3$, $P = 0.0018$) followed by Bonferroni-Holm post-hoc test correction. **f** Left: Schema of passive contact. Middle: PETH of normalized ΔF/F for SC-VTA-projecting neurons, centered on passive contacts. Right: Mean ΔF/F (Z-score) before, during and after passive contacts. RM one-way ANOVA (Events main effect: $F_{(2,7)} = 2.15$, $P = 0.1528$). **g** Left: Schema of ipsilateral recorded orientation to distal social stimulus. Middle: Peri-event time histogram (PETH) of normalized ΔF/F for SC-VTA-projecting neurons, centered on distal ipsilateral orientation. Right: Mean ΔF/F (Z-score) before, during and after distal ipsilateral orientation. RM one-way ANOVA (Events main effect: $F_{(2,7)} = 3.8947$, $P = 0.0452$) followed by Bonferroni-Holm post-hoc test correction. **h** Left: Schema of ipsilateral recorded orientation to proximal social stimulus. Middle: Peri-event time histogram (PETH) of normalized ΔF/F for SC-VTA-projecting neurons, centered on proximal ipsilateral orientation. Right: Mean ΔF/F (Z-score) before, during and after proximal ipsilateral orientation. RM one-way ANOVA (Events main effect: $F_{(2,7)} = 14.45$, $P = 0.0004$) followed by Bonferroni-Holm post-hoc test correction. **i** Left: Schema of passive crossing of social stimulus in the frontal field. Middle: Peri-event time histogram (PETH) of normalized ΔF/F for SC-VTA-projecting neurons, centered on passive crossing. Right: Mean ΔF/F (Z-score) before, during and after passive crossing. RM one-way ANOVA (Events main effect: $F_{(2,7)} = 1.3413$, $P = 0.2931$) followed by Bonferroni-Holm post-hoc test correction. **j** Left: Schema of contralateral recorded orientation to proximal social stimulus. Middle: Peri-event time histogram (PETH) of normalized ΔF/F for SC-VTA-projecting neurons, centered on proximal contralateral orientation. Right: Mean ΔF/F (Z-score) before, during and after proximal contralateral orientation. RM one-way ANOVA (Events main effect: $F_{(2,7)} = 0.3474$, $P = 0.7125$) followed by Bonferroni-Holm post-hoc test correction. **k** Left: Schema of contralateral recorded orientation to distal social stimulus. Middle: Peri-event time histogram (PETH) of normalized ΔF/F for SC-VTA-projecting neurons, centered on distal contralateral orientation. Right: Mean ΔF/F (Z-score) before, during and after distal contralateral orientation. RM one-way ANOVA (Events main effect: $F_{(2,7)} = 0.1090$, $P = 0.8975$) followed by Bonferroni-Holm post-hoc test correction. N indicates the number of mice. All the data are shown as the mean ± s.e.m. as error bars or error bands. Source data are provided as a Source Data file.

**SC-VTA pathway optostimulation increases exploratory behavior and does not induce place preference**. Since mice are less oriented towards a salient stimulus during SC-VTA pathway stimulation, we hypothesized that manipulation of SC-VTA pathway would increase locomotion. In order to verify our hypothesis, eYFP- and ChR2-expressing mice were placed in an open field for 10 min (Fig. 6a–c). Although we did not observe significant differences between light OFF and light ON epochs using a within-group analysis, ChR2-expressing mice increased the distance moved relative to control animals upon SC terminal stimulation (Fig. 6d). To exclude anxiolytic effects induced by SC-VTA pathway stimulation, we measured the time spent in the center of the arena, which was similar between eYFP- and ChR2-expressing mice (Fig. 6e). We also observed that following single burst optogenetic stimulation, ChR2-expressing mice showed an increased number of head/body turns compared to controls (Fig. 6f).

Finally, we tested whether manipulation of SC-VTA pathway supports real-time place preference. Animals were placed into a two-chambered arena, where only one chamber was paired with optogenetic stimulation or inhibition (Supplementary Fig. 5a). eYFP-control, ChR2-expressing and Jaws-expressing mice spent a comparable amount of time in each chamber (Supplementary Fig. 5b).

**The SC projects to VTA DAT$^+$ neurons connected with dorsolateral striatum (DLS)**. Since the evident functional divergence between the SC- or the mPFC-VTA pathways, we hypothesized that SC targets a neuronal subpopulation that is anatomically different from the one targeted by the mPFC. Midbrain DAT$^+$ neuron axons innervate the striatum with a gradient distribution of cells projecting to the ventromedial central and dorsolateral striatum[20]. We, therefore, injected Cholera Toxin Subunit-B (CTB)−488 in the Nucleus Accumbens (NAc) and CTB-555 in the dorsolateral striatum (DLS) and we imaged the VTA (Fig. 7a–c). As previously reported[21], we found that VTA neurons projecting to the NAc and to the DLS segregate in two largely

non-overlapping neuronal populations (Fig. 7d). Furthermore, we used optogenetics combined with retrograde tracing to investigate which subpopulation of VTA DAT$^+$ neurons is functionally connected with the SC. We injected retrograde AAVrg-pCAG-FLEX-tdTomato in either DLS or NAc in DAT-Cre mice to label distinct VTA DAT$^+$ projecting neurons (TH staining on post-hoc slices in order to verify the neuronal subtype; Fig. 7e, f). Within the same animals, we injected AAV5-hSyn-ChR2-eYFP in the SC to assess biases in input connectivity based on output-specificity. We performed whole-cell patch-clamp recordings from identified VTA DAT$^+$ projecting neurons and found that SC makes functional excitatory synapses with VTA DAT$^+$ neurons projecting to the DLS (46.42%), while VTA DAT$^+$-NAc are poorly connected with the SC cells (Fig. 7g–i). As previously shown, the VTA DAT$^+$ neurons connected to SC are located in a very lateral portion of the VTA (Fig. 7j). Finally, since it has been shown that mPFC makes monosynaptic inputs onto NAc-projecting VTA DAT$^+$ neurons[8], we hypothesized that VTA DAT$^+$-DLS and VTA DAT$^+$-NAc pathways play different roles during conspecific interaction. In particular, we hypothesized that photostimulation of the VTA DAT$^+$-DLS pathway would mimic the effects of SC-VTA stimulation. To test this idea, we injected Cre-dependent ChR2 (AAV5-EF1α-DIO-ChR2-eYFP) or Cre-dependent eYFP (AAV5-Ef1α-DIO-eYFP) in the VTA of DAT-Cre mice, and we implanted a fiber optic in either DLS or NAc (Fig. 7k–l). Experimental mice underwent the same behavioral protocol of free social interaction and optogenetic stimulation described previously (Fig. 5b, c). As expected, photostimulation of VTA DAT$^+$–NAc pathway in ChR2-expressing mice increased time interaction between light-OFF and light-ON conditions while no changes were observed in eYFP-control group (Fig. 7m)[4]. Remarkably, photostimulation of VTA DAT$^+$-DLS in ChR2-expressing mice decreases time sniffing the unfamiliar conspecific (Fig. 7m), recapitulating the effects of SC-VTA pathway stimulation. Interestingly, differences in time spent in interaction with a conspecific in VTA DAT$^+$-NAc and VTA DAT$^+$-DLS mice occur during the first minute of interaction,

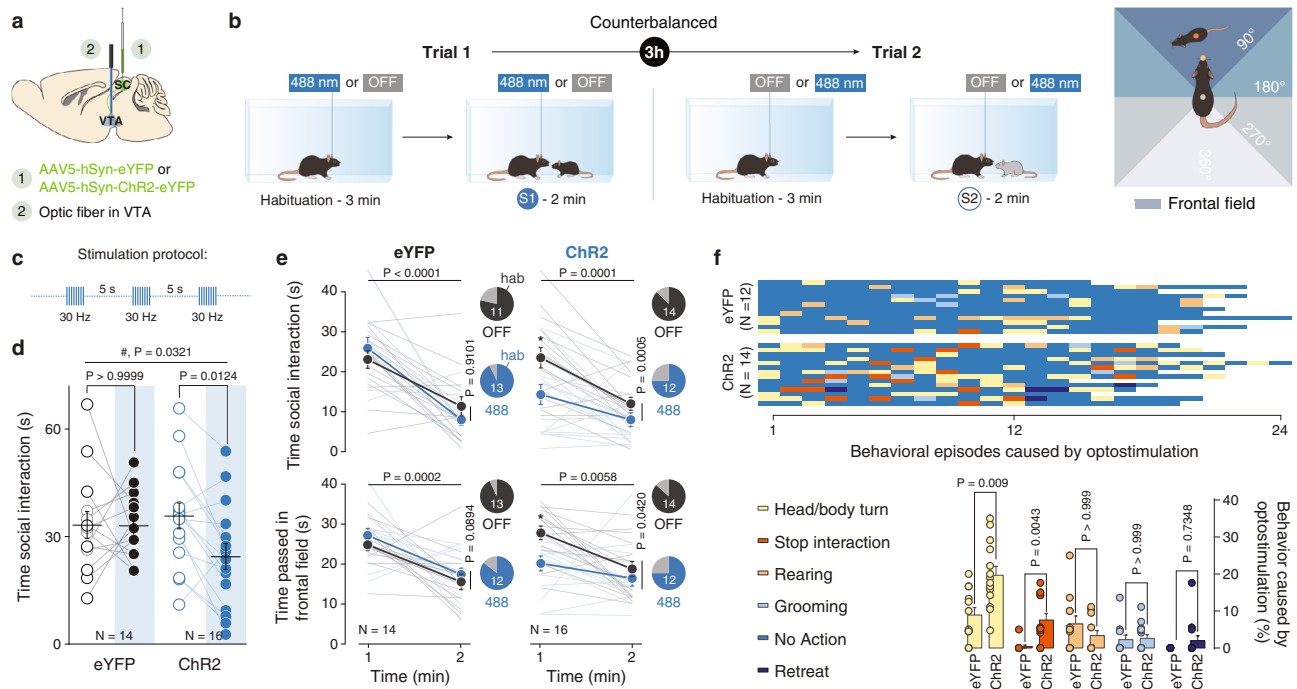

**Fig. 5 Optogenetic manipulation of SC-VTA pathway alters social interaction and perturbs head orientation towards conspecific. a** Schema of injections sites in SC with AAV5-hSyn-eYFP or AAV5-hSyn-ChR2-eYFP, and optic fiber implantation above the VTA. **b** Left panel: Schema of free social interaction. The eYFP- and ChR2-expressing mice freely interacted with two different unfamiliar mice under both stimulation conditions. Right panel: schema representing the position of the frontal field. **c** Stimulation protocols: 8 pulses of 488 nm light (30 Hz) were separated by 5 s in the light ON condition. **d** Time social interaction during the free social interaction test for eYFP- and ChR2-expressing mice in the SC. RM two-way ANOVA (Light main effect: $F_{(1,28)} = 5.0855$, $P = 0.0321$; Virus main effect: $F_{(1,28)} = 0.8528$, $P = 0.3637$; Light × Virus Interaction: $F_{(1,28)} = 4.1962$, $P = 0.0500$) followed by Bonferroni-Holm post-hoc test correction. **e** Upper panels: time passed interacting with the social stimulus for the 1st and 2nd minute of the free social interaction test in light and no-light conditions. RM two-way ANOVA (eYFP: Light main effect: $F_{(1,13)} = 0.013$, $P = 0.9101$; Time main effect: $F_{(1,13)} = 34.64$, $P < 0.0001$; Light × Time Interaction: $F_{(1,13)} = 3.487$, $P = 0.085$. ChR2: Light main effect: $F_{(1,15)} = 19.43$, $P = 0.0005$; Time main effect: $F_{(1,15)} = 26.38$, $P = 0.0001$; Light × Time Interaction: $F_{(1,15)} = 1.878$, $P = 0.1907$) followed by Bonferroni's multiple comparisons post-hoc test. Lower panels: time passed with the social stimulus in the frontal field for the 1st and 2nd minute of the free social interaction test in light and no-light conditions. RM two-way ANOVA (eYFP: Light main effect: $F_{(1,13)} = 3.368$, $P = 0.0894$; Time main effect: $F_{(1,13)} = 25.31$, $P = 0.0002$; Light × Time Interaction: $F_{(1,13)} = 0.01355$, $P = 0.9091$. ChR2: Light main effect: $F_{(1,15)} = 5.006$, $P = 0.042$; Time main effect: $F_{(1,15)} = 10.59$, $P = 0.0058$; Light × Time Interaction: $F_{(1,15)} = 2.749$, $P = 0.1196$) followed by Bonferroni's multiple comparisons post-hoc test. Pie charts represent the percentage of mouse that decreases the interaction/orientation between 1st and 2nd minute. **f** Upper panel: identification and report of the behavioral episodes caused by a burst of optostimulation. Lower panel: quantification of the behavioral episodes caused by a burst of optostimulation. Two-way ANOVA (Behavior main effect: $F_{(2.791, 66.99)} = 23.43$, $P < 0.0001$; Group main effect: $F_{(1, 24)} = 12.73$, $P = 0.0016$; Behavior × Group Interaction: $F_{(4, 46)} = 6.539$, $P = 0.0001$) followed by Bonferroni's multiple comparisons post-hoc test. N indicates the number of mice. # indicates significantly different interaction. All the data are shown as the mean ± s.e.m. as error bars. Source data are provided as a Source Data file.

while animals habituate to the interaction during the second minute (Supplementary Fig. 6a, b). However, neither VTA DAT⁺-DLS nor VTA DAT⁺-NAc animals showed differences in the orientation of the head (Supplementary Fig. 6a, b). Altogether these data indicate that optogenetic modulation of VTA DAT⁺-NAc and VTA DAT⁺-DLS oppositely regulates social interaction while does not per se change orienting responses.

## Discussion

The present study examines the role of the SC-VTA pathway during unfamiliar conspecific interaction. We parsed the involvement of two different VTA pathways in distinct, but not mutually exclusive, components of social interaction: while SC-VTA pathway encodes orienting response towards an unfamiliar conspecific, mPFC-VTA circuit is activated during the investigation of the social stimuli.

It is well established that individuals exhibit orienting responses toward novel, unexpected and salient environmental stimuli[6]. Once the salient stimulus is detected, its identity and its

potential behavioral relevance need to be considered to decide whether to approach or avoid it. In the context of social behavior, an individual needs to detect the presence of other conspecifics and then estimate the valence of a hypothetical interaction, in order to make decisions on whether to initiate or avoid the contacts. After the first interaction, an individual updates the neuronal representation of the value associated with that interaction to guide its subsequent decisions[22] to or -not- to maintain that social contact. Although it has been previously suggested that deficits in orienting response may underlie some of the social behavior deficits associated with Autism Spectrum Disorders[23], the importance of orienting responses in social interaction has been largely ignored. Here we confirmed that VTA plays an important role during social interaction[24] and we demonstrated that different inputs to this region convey different signals. Among these inputs, we showed that SC-VTA pathway encodes orienting response toward conspecific, and alterations of its activity affects social interaction. Importantly, our data indicate that SC to VTA pathway is activated also during orienting

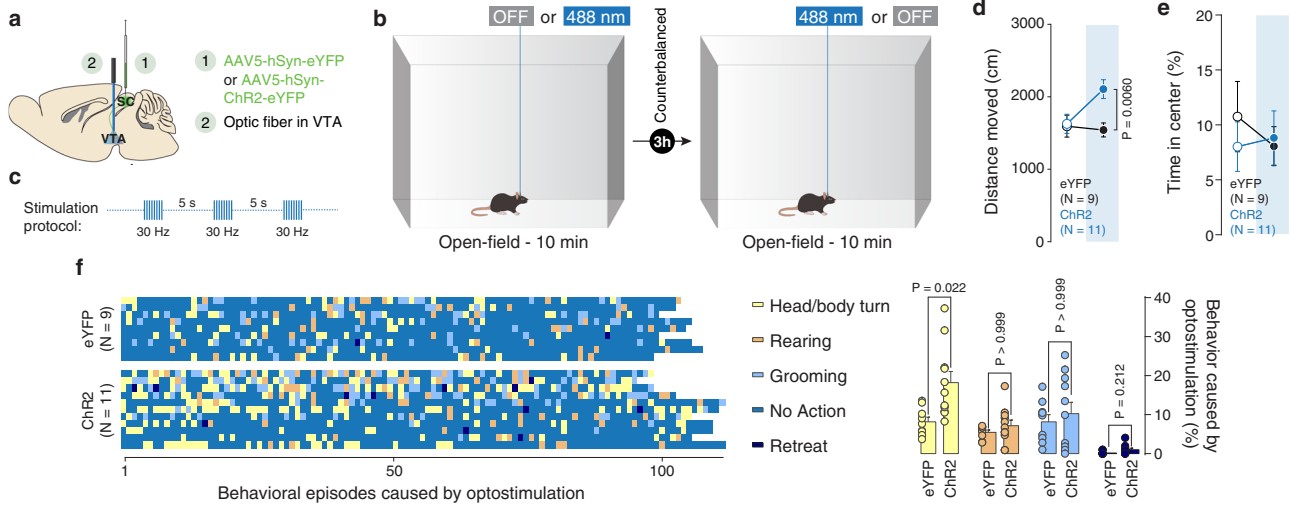

**Fig. 6 SC-VTA optostimulation increases exploratory behavior. a** Schema of injections sites in SC with AAV5-hSyn-eYFP or AAV5-hSyn-ChR2-eYFP, and optic fiber implantation above the VTA. **b** Schema of the open-field arena. The mice are free to explore the apparatus for 10 min in both stimulations' conditions. **c** Stimulation protocols: 8 pulses of 488 nm light (30 Hz) were separated by 5 s in the light ON condition. **d** Distance moved in the open-field arena for eYFP and ChR2 mice under light and no-light stimulation. RM two-way ANOVA (Light main effect: $F_{(1,18)} = 4.0046$, $P = 0.0607$; Virus main effect: $F_{(1,18)} = 4.8210$, $P = 0.0415$; Light × Virus interaction: $F_{(1,18)} = 4.8217$, $P = 0.0415$) followed by Bonferroni-Holm post-hoc test correction. **e** Time passed in the center of the open-field arena for eYFP and ChR2 mice under light and no-light stimulation. RM two-way ANOVA (Light main effect: $F_{(1,18)} = 0.1628$, $P = 0.6913$; Virus main effect: $F_{(1,18)} = 0.145$, $P = 0.7078$; Light × Virus interaction: $F_{(1,18)} = 0.546$, $P = 0.4695$). **f** Right panel: identification and report of the behavioral episodes caused by a burst of optostimulation. Left panel: quantification of the behavioral episodes caused by optostimulation. Two-way ANOVA (Behavior main effect: $F_{(2.004, 36.08)} = 20.47$, $P < 0.0001$; Group main effect: $F_{(1, 18)} = 5.370$, $P = 0.0325$; Behavior × Group Interaction: $F_{(3, 54)} = 3.364$, $P = 0.0251$) followed by Bonferroni's multiple comparisons post-hoc test. N indicates the number of mice. # indicates significantly different interaction. All the data are shown as the mean ± s.e.m. as error bars. Source data are provided as a Source Data file.

response toward a moving ball, suggesting that more generally, the activity of this circuit encode orientation toward salient stimuli.

SC is an evolutionarily ancient structure organized in functionally and anatomically distinct layers. While the upper layers are exclusively visual, the medial and deeper layers have multi-sensory and motor functions[25]. It has been suggested that this structure, via different output projections, mediates both orienting and avoidance responses to novel sensory stimuli[9,11]. Moreover, SC lesions generally result in deficits in visual orientation, sensory neglect[26] and aberrant predatory responses[27]. Previous reports have shown that SC mediates visually-induced defensive behaviors. Indeed, the activity of excitatory neurons in the deep layers of the medial SC represents the saliency of the threat stimulus and is predictive of escape behavior[28,29]. More specifically, the SC projections onto the VTA GABA neurons would promote "flight" behavior in threatening context[19]. Here, we demonstrated, for the first time, the importance of the SC in social contexts. Indeed, our data show that SC-VTA pathway is activated during orientation toward an unfamiliar salient stimulus and returns to baseline upon social contact. Perturbing the physiological activity pattern of this pathway affects orientation and, consequently, social interaction. Importantly, while photoactivation of SC to VTA pathway alters orientation and decreases social interaction, photoinhibition of the same pathway did not induce a diametrically opposite behavior. This apparent inconsistency could be explained by the differences in stimulation protocols. Indeed, while light inhibition is continuous and constant during the task, photo excitation occurs every 5 seconds with a 30 Hz burst of 8 pulses of light. Furthermore, it is important to mention that all the experiments of this study were performed during the light cycle when normal mice are less active[30]. Since the activity of SC neurons varies during mouse wakefulness[21,31], further studies will need to investigate

how the activity of SC-VTA pathway changes based on the circadian rhythm. Taken together our findings are consistent with the hypothesis that orienting responses require a tightly tuning of activity of SC-VTA pathway to environmental stimuli. Importantly, although we worked only with male mice, we hypothesize that similar mechanisms could also be observed in female C57Bl/6 J.

In this paper, we showed that neurons located in the intermediate and deep layers of the SC project to the VTA and form monosynaptic connections with both VTA DA and GABA neurons. We cannot exclude that the connections between SC and VTA are partly due to collaterals coming from fibers en passant travelling to downstream target structures heading to oculomotor centers, caudal pons or medulla oblongata. Previous studies have demonstrated a similar phenomenon in the case of SNc, in which crossed projections arising from the intermediate layer of SC targeting specifically the contralateral pontomedullary reticular formation (PMRF) give off collaterals onto SNc neurons[32]. Nevertheless, we demonstrated that dopaminergic neurons receiving inputs from the SC are mainly projecting to the DLS and that VTA$^{DAT+}$-DLS neurons control social interaction. As a consequence, these findings suggest that the SC projections onto the VTA are upstream of social interaction.

Through patch-clamp experiments we demonstrated that SC forms functional synaptic connections with both VTA-DAT$^+$ and -GAD$^+$ neurons. Recently, it has been shown that GABA neurons in the VTA would encode head angles for each of the three principal axes of rotation, and it has been suggested that the control of head rotation is essential for most motivated behaviors[33]. Here we report that SC-VTA pathway is activated during orienting response toward a salient stimulus and regulates social behavior. Focusing on VTA DA neurons, we found that the ones projecting to DLS are controlled by SC and that their stimulation decreases social interaction. Interestingly, it has been

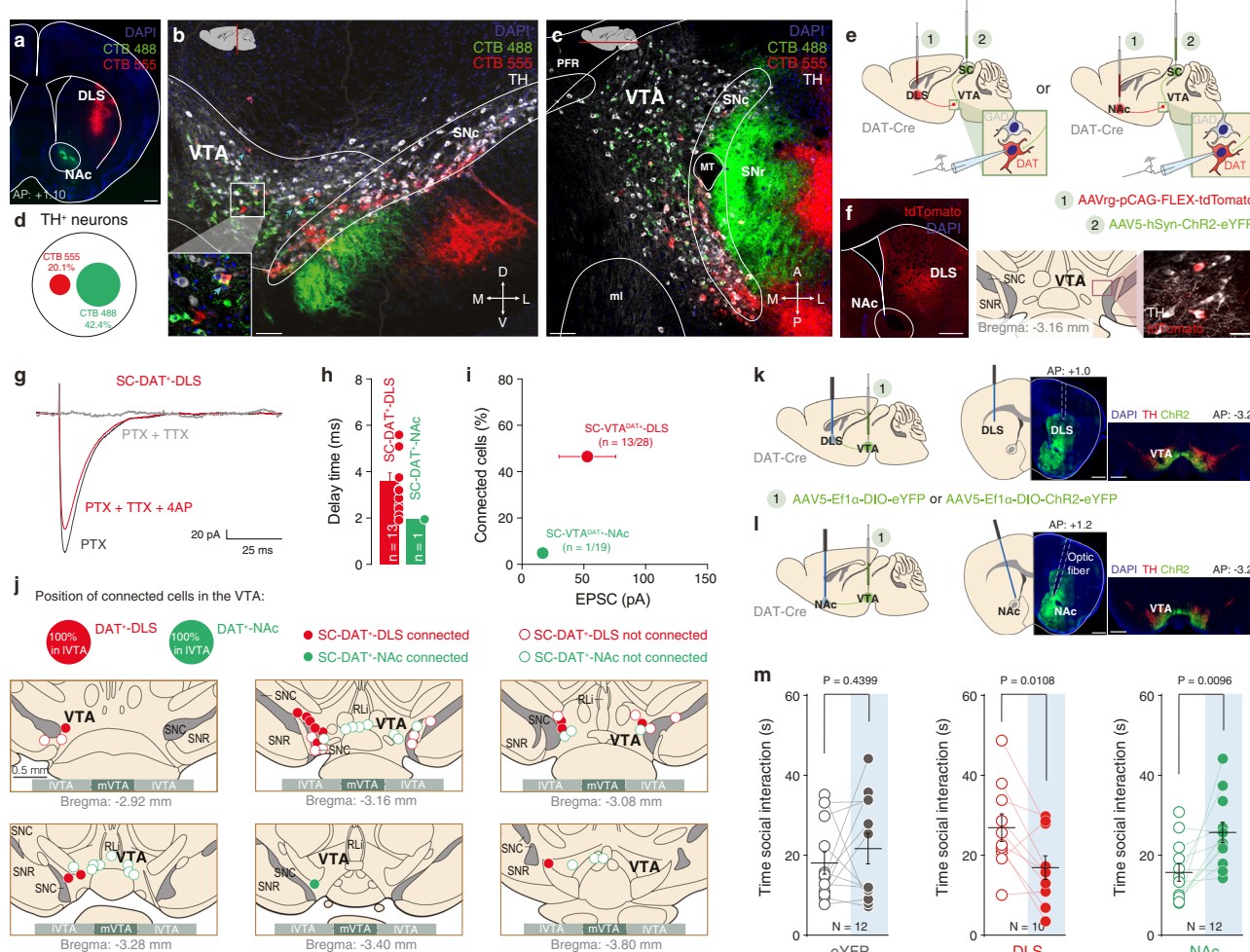

**Fig. 7 SC is mainly connected to VTA DAT+ neurons which project to DLS. a** Representative image of the NAc and DLS in coronal slice infected with the CTB-488 and CTB-555 (scale bar 500 μm. This experiment was reproduced at least three times). **b** Coronal plane image of the VTA with TH staining (white) and infected cells projecting to the NAc or DLS (scale bar = 100 μm. This experiment was reproduced at least three times). **c** Horizontal plane image of the VTA with TH staining (white) and infected cells projecting to the NAc or DLS (scale bar = 100 μm. This experiment was reproduced at least three times). **d** Proportion of TH+ and either TH+/CTB-488+ (NAc-projecting VTA DAT+ neurons) or TH+/CTB-555+ (DLS-projecting VTA DAT+ neurons). **e** Schema of injections in DAT-Cre mice. The SC was infected with AAV5-hSyn-ChR2-eYFP and the NAc or the DLS with the AAVrg-pCAG-FLEX-tdTomato. Whole-cell patch was then performed in DAT+ neurons projecting to two different regions. **f** Left panel: Representative image of the AAVrg-pCAG-FLEX-tdTomato site of injection in the DLS (scale bar = 500 μm). Right panel: example of DAT+ neurons projecting to DLS (scale bar = 20 μm). This experiment was reproduced at least three times. **g** Example traces of optogenetically elicited excitatory postsynaptic currents (EPSCs) in VTA DAT+ neurons projecting to DLS in presence of PTX, TTX, and 4AP. **h** Delay time of the EPSCs for VTA DAT+-NAc or DAT+-DLS neurons. **i** Quantification of connected cells from SC onto VTA DAT+-NAc or DAT+-DLS neurons in relation of the amplitude of EPSCs. The VTA DAT neurons receiving projections from the SC are mainly projecting to the DLS with the highest current amplitude. **j** Position of some patched VTA DAT+-NAc or DAT+-DLS neurons. SC-VTA^DAT+-DLS connected neurons are mainly in the lateral part of the VTA (lVTA). **k** Left panel: Schema of injections sites in the VTA of DAT-Cre mice with AAV5-Ef1α-DIO-eYFP or AAV5-Ef1α-DIO-ChR2-eYFP, and optic fiber implantation above the DLS. Middle panel: representative image showing the fiber optic's track in the DLS (scale bar = 500 μm). Right panel: representative image of the site of injection in the VTA (scale bar = 500 μm). Similar viral expression and OF location were observed in all the mice that performed the experiment which results are shown in Fig. 7l. **l** Left panel: Schema of injections sites in the VTA of DAT-Cre mice with AAV5-Ef1α-DIO-ChR2-eYFP or AAV5-Ef1α-DIO-eYFP, and optic fiber implantation above the NAc. Middle panel: representative image showing the fiber optic's track in the NAc (scale bar = 500 μm). Right panel: representative image of the site of injection in the VTA (scale bar = 500 μm). Similar viral expression and OF location were observed in all the mice that performed the experiment which results are shown in Fig. 7l. **m** Left panel: VTA DAT+-NAc^eYFP and VTA DAT+-DLS^eYFP mice do not change time of social interaction between the two stimulation conditions. Paired t-test two-sided ($t_{(11)} = -0.8013$). Middle panel: VTA DAT+-DLS^ChR2 mice decrease the time of social interaction during light ON condition. Paired t-test two-sided ($t_{(11)} = 3.2015$). Right panel: VTA DAT+-NAc^ChR2 mice increase the time of social interaction during light ON condition. Paired t-test two-sided ($t_{(11)} = -2.8177$). N, n indicate number of mice and cells respectively. All the data are shown as the mean ± s.e.m. as error bars. Source data are provided as a Source Data file.

suggested that the looped circuit connecting the SC to the basal ganglia may be a "candidate mechanism to perform the pre-attentive selections required to determine whether gaze should be shifted, and if so, to which stimulus"[12].

Remarkably and contrary to the data presented in Fig. 5 (stimulation of SC to VTA pathway), stimulation of the VTA to DLS pathway does not affect the time passed in the frontal field between OFF and ON conditions. The apparent inconsistency

could be explained by the fact that when we stimulated SC to VTA pathway, we activated both DA and GABA neurons. Although our results suggest that DA neurons projecting to DLS are per se not involved in head rotation and orienting response, putting together, these data suggest that SC to VTA triggers orienting behavior that is essential for downstream-regulated motivated behaviors.

Social behavior involves a sequence of different actions: from interruption of the ongoing behavior, stimulus evaluation, decision-making and further exploration. In particular, social motivation is characterized by preferential orientation towards social stimuli, evaluation of the rewarding properties of conspecific interaction and effort to maintain social bonds[24]. Social motivation deficits play a central role in Autism Spectrum Disorders (ASDs), neuropathology characterized by major social behavior alterations. Although evidence suggests that VTA is central to social deficits in ASD patients[34], the origins of these deficits are still largely unknown. Interestingly, one of the core diagnostic criteria for ASD includes a decrease in eye-contact[35] and eye-tracking experiments have shown impairment in orientation towards social stimuli in ASD patients[36]. Here, our data support the hypothesis that a precise activation of inputs to the VTA is necessary to promote social orientation and then interaction. Indeed, by using ChR2 within the SC-VTA pathway, we disrupt the time-lock increase in activity during orientation towards conspecifics. This manipulation per se is sufficient to result in altered social interaction. We, therefore, propose that intact functionality of the SC-VTA pathway may be fundamental for orientation towards conspecifics and deficits in this pathway may participate in social motivation dysfunctions in ASD patients.

In conclusion, our data strongly suggest that while SC-VTA pathway encodes orienting response towards an unfamiliar conspecific or moving stimuli, mPFC-VTA circuit function regulates the investigation and maintenance of social interaction. Elucidating the brain circuits underlying conspecific interaction is therefore essential, not only to understand how social behavior occurs but also to comprehend the aberrant neural mechanisms underlying social deficits in psychiatric disorders.

## Methods

**Mice**. Adult (8–16 weeks old) male wild type (WT; C57Bl/6 J), DAT-iresCre (Slc6a3$^{tm1.1(cre)Bkmn}$/J, called DAT-Cre in the rest of manuscript) and GAD2-iresCre (Gad2$^{tm2(creZjh)}$/J, called GAD65-Cre in the rest of manuscript) mice were employed for this study. Mice were housed in groups (weaning at P21-P23) under a 12 h light-dark cycle (7:00 a.m.–7:00 p.m.) with food and water ad libidum. All physiology and behavior experiments were performed during the light cycle (the experiments were performed in a time window that started ~2 h after the end of the dark cycle and ended 2 h before the start of the next dark cycle) and were conducted in a room with fixed illumination (20 Lux), temperature (22–24 °C), and humidity (50%). All the procedures performed at UNIGE complied with the Swiss National Institutional Guidelines on Animal Experimentation and were approved by the respective Swiss Cantonal Veterinary Office Committees for Animal Experimentation.

### Stereotaxic injection and optic fiber implantation

*Optogenetic experiments.* rAAV5-hSyn-eYFP (control), rAAV5-hSyn-hChR2(H134R)-eYFP or rAAV5-hSyn-Jaws-KGC-GFP-ER2 were injected in WT mice at 8 weeks. Mice were anesthetized with a mixture of oxygen (1 L/min) and isoflurane 3% (Baxter AG, Vienna, Austria). The skin was shaved, locally anesthetized with 40–50 μL lidocaine 0.5% and disinfected. The animals were placed in a stereotactic frame (Angle One; Leica, Germany) and bilateral injections were performed in the SC (ML ± 0.8 mm, AP − 3.4 mm, DV − 1.5 mm from Bregma, 500 nL per side). The virus was injected via a glass micropipette (Drummond Scientific Company, Broomall, PA). The virus was incubated for 3–4 weeks and subsequently mice were implanted with an optic fiber above the VTA with the following protocol. The animals were anesthetized, placed in a stereotactic frame, the skin was shaved, and a unilateral craniotomy was performed above the VTA. The optic fiber (∅ 200 μm) was implanted with a 10° angle at the following coordinates: ML ± 0.9 mm, AP − 3.2 mm, DV − 3.95 mm from Bregma above the VTA and fixed to the skull with dental acrylic. For optogenetic experiments using

DAT-Cre mice, the animals were injected using the protocol described above with rAAV5-Ef1α-DIO-eYFP or rAAV5-Ef1α-DIO-hChR2(H134R)-eYFP in the VTA (ML ± 0.5 mm, AP − 3.2 mm, DV − 4.25/−4.00 mm from Bregma). The virus was incubated for 3–4 weeks and subsequently mice were implanted with optic fibers above either the Nucleus Accumbens (NAc) with 15° angle at the following coordinates: ML ± 2.0 mm, AP + 1.2 mm, DV − 4.2 mm from Bregma; or the dorsolateral Striatum (DLS) at the following coordinates: ML ± 2.0 mm, AP + 1.0 mm, DV − 2.5 mm from Bregma. Injections and implantation sites were confirmed post-hoc.

*Fiber photometry experiments.* AAVrg-Ef1α-mCherry-IRES-Cre injections in the VTA (500 nL in total) and unilateral AAV9-hSyn-FLEX-GCaMP6s-WPRE-SV40 injection unilaterally in the SC (500 nL: ML + or −0.8 mm, AP −3.4 mm, DV − 1.5 mm from Bregma) or mPFC (500 nL: ML + or −0.3 mm, AP + 1.95 mm, DV − 2.1 mm from Bregma) in adult WT mice were conducted following the protocol described above. The viruses were incubated 3–4 weeks prior to optic fibers implantation. The optic fibers (∅ 200 μm) were unilaterally implanted following the above-mentioned protocol in the SC (ML + or −0.8 mm, AP −3.4 mm, DV − 1.5 mm from Bregma) and mPFC (ML + or −0.3 mm, AP + 1.95 mm, DV − 2.1 mm from Bregma). The mice prepared for fiber photometry experiments performed both behavioral tests: the free social interaction and orientation test. Injections and implantation sites were confirmed post-hoc.

*Ex vivo electrophysiological recording experiments.* Injections of rAAV5-hSyn-DIO-mCherry, AAVrg-pCAG-FLEX-tdTomato-WPRE and/or rAAV5-hSyn-hChR2(H134R)-eYFP, were performed in WT, DAT-Cre or GAD65-Cre mice (depending on the experiment). rAAV5-hSyn-hChR2(H134R)-eYFP was injected in the SC, at the same coordinates previously described. rAAV5-hSyn-DIO-mCherry was injected in the VTA at the same coordinates previously described. AAVrg-pCAG-FLEX-TdTomato was bilaterally injected either in the NAc at these coordinates: ML ± 1.0 mm, AP + 1.2 mm, DV − 4.4 / −4.0 mm from Bregma (500 nL each side); either in the DLS at following coordinates: ML ± 2.0 mm, AP + 1.0 mm, DV − 2.8 mm from Bregma (500 nL each side). The viruses were incubated 3–4 weeks before to perform ex vivo electrophysiological recordings.

*Anatomical validation experiments.* Bilateral injections of rAAV5-hSyn-eYFP (500 nL per side) in the SC were performed as described above. Unilateral injection of ssAAV-retro/2-hSyn1-chl-tdTomato-WPRE-SV40p(A) (200 nL) in the VTA were performed as described above. The retrograde experiment was repeated with a second batch of animals injected with a unilateral CAV-cre injection in the VTA (500 nL) and a rAAV5-hSyn-DIO-mCherry injection in the SC (500 nL per side).

WT mice were bilaterally injected using Cholera Toxin subunit-B Alexa fluor 555 (CTB-555) or CTB-488, respectively, in the DLS (ML ± 2.0 mm, AP + 1.0 mm, DV − 2.8 mm from Bregma, 200 nL each side) and the NAc (ML ± 1.0 mm, AP + 1.2 mm, DV − 4.4 / −4.0 mm from Bregma, 200 nL each side). The CTB-488 and CTB-555 were incubated for 10 days before immunostaining procedures.

### Social behavior tests

*Social and non-social orientation test during in vivo calcium imaging.* The arena of the orientation test consists of two cylinders (height = 25 cm) positioned one inside the other. The smaller cylinder (∅ = 8 cm) is composed of transparent plastic and presents small holes (∅ = 0.3 cm) which prevent social contact but allow olfactory, auditory and visual cues. The mice injected with AAVrg-Ef1α-mCherry-IRES-Cre in the VTA and AAV9-hSyn-FLEX-GCaMP6s-WPRE-SV40 in the SC or mPFC were gently placed in the small cylinder, which allow the experimental animal only to rotate. After a 5 min period of habituation, a social stimulus (unfamiliar juvenile conspecific sex-matched C57BL/6 J, 3–4 weeks) or a moving ball (64.60, Sphero Mini, Sphero Edu) was introduced between the small and the big cylinder (∅ = 20 cm) in the external circular corridor. The stimulus or the moving ball were allowed to freely move in the intercylinder space for 5 min. The ball was programmed to move similarly to a conspecific, with approximately the same speed. During the entire session the bulk calcium waves of SC- or mPFC-VTA-projecting neurons were recorded with a 1-site fiber photometry system (Doric lenses).

*Orientation test during optogenetic manipulation.* The arena described previously was used during optogenetic experiments. All eYFP-, ChR2- and Jaws-expressing mice underwent both conditions of light-ON or light-OFF epochs. The light activation or inhibition protocols were applied depending on a random assignment and lasted during all the trials. For eYFP-control and ChR2-expressing mice, the following optogenetic stimulation protocol was used: burst of 8 pulses of 4 msec light at 30 Hz every 5 sec (Imetronic, Pessac, France), wavelength of 488 nm (BioRay Laser, Coherent). For eYFP-control and Jaws-expressing mice, a constant light-ON epoch was delivered (Imetronic, Pessac, France), at 640 nm wavelength (BioRay Laser, Coherent). The experimental mice were gently placed in the internal cylinder. After a 5 min period of habituation, a social stimulus (unfamiliar juvenile conspecific sex-matched C57BL/6 J, 3–4 weeks) or a moving ball (64.60, Sphero Mini, Sphero Edu) was introduced between the small and the big cylinder (∅ = 20 cm) in the external circular corridor. The stimulus or the moving ball were allowed to freely move in the intercylinder space for 2 min. The ball was

programmed to move similarly to a conspecific, with approximately the same speed. The mice were then removed and placed in their respective cages. After 3 h, the same experimental design was performed and the mice that received the light stimulation protocol did not receive it and vice versa. Another unfamiliar conspecific stimulus (S2) or moving ball (mO) was introduced for 2 min in the intercylinder space after the 5 min period of habituation. The power expected at the tip of the optic fiber was between 8-12 mW. To this aim the laser power was checked before every experiment and the power at the optic fiber tip was controlled before any implantation.

*Free interaction during in vivo calcium imaging.* An arena similar to the animal's homecage was used for the free social interaction test. The arena was cleaned using 70% ethanol and the bedding replaced after each trial. Mice injected with AAVrg-Ef1α-mCherry-IRES-Cre in the VTA and AAV9-hSyn-FLEX-GCaMP6s-WPRE-SV40 in the SC or mPFC were first placed in the arena and were free to explore the new environment for 5 min. After this habituation period, a social stimulus (unfamiliar juvenile conspecific sex-matched C57BL/6 J, 3–4 weeks) was introduced in the cage, and the animals were free to interact for 5 min. During the entire session the bulk calcium waves of SC- or mPFC-VTA-projecting neurons were recorded with a 1-site fiber photometry system (Doric lenses).

*Free interaction during optogenetic manipulation.* Equally to the protocol described previously, an arena similar to the animal's homecage was used for the free social interaction test. The arena was cleaned using 70% ethanol and the bedding replaced after each trial. All eYFP-, ChR2- and Jaws-expressing mice underwent both conditions of light-ON or light-OFF epochs. The light stimulation or inhibition protocols were applied depending on a random assignment and lasted during all the trials. For eYFP-control and ChR2-expressing mice, the following optogenetic stimulation protocol was used: burst of 8 pulses of 4 msec light at 30 Hz every 5 sec (Imetronic, Pessac, France), wavelength of 488 nm (BioRay Laser, Coherent). For eYFP-control and Jaws-expressing mice, a constant light-ON epoch was delivered (Imetronic, Pessac, France), at 640 nm wavelength (BioRay Laser, Coherent). The experimental mice were first placed in the arena and were free to explore the new environment for 3 min. After this period, a social stimulus (unfamiliar juvenile conspecific sex-matched C57BL/6 J, 3–4 weeks; S1) was introduced in the cage, and the animals were free to interact for 2 min. The mice were then removed and placed in their respective cages. After 3 h, the same experimental design was performed and the mice that received the light stimulation protocol did not receive it and vice versa. Another unfamiliar conspecific stimulus (S2) was introduced for 2 min in the cage after the 3 min of exploration during the second trial. The power expected at the tip of the optic fiber was between 8 and 12 mW. To this aim the laser power was checked before every experiment and the power at the optic fiber tip was controlled before any implantation.

Likewise, DAT-Cre mice injected with rAAV5-Ef1α-DIO-eYFP or rAAV5-Ef1α-DIO-hChR2(H134R)-eYFP in the VTA and implanted with double optic fiber either in the Nucleus Accumbens (VTA DAT+-NAc^eYFP or VTA DAT+-NAc^ChR2) or Dorsolateral Striatum (VTA DAT+-DLS^eYFP or VTA DAT+-DLS^ChR2) underwent the same protocol of free social interaction test as described above.

The arena was cleaned using 70% ethanol between each trial. Every session of every social behavior test was video-tracked and recorded using Ethovision XT (Noldus, Wageningen, the Netherlands). Using the tracking of the body parts of the experimental mice and the stimuli, the distance between them was automatically calculated. Moreover, thanks to the vector formed by the gravity center and the nose of the experimental mice, it has been possible to calculate the head orientation ($\omega$) towards the gravity center of the stimulus and its relative position. $\omega(t) = 0$ events were identified across the sessions and according to the recorded hemisphere, the movements performed by the experimental mice and the distance between animals, the events were separated in passive crossing, ipsi- or contra-recorded orientation proximal or distal to the stimulus. A $\omega(t) = 0$ event was taken into consideration for the final analysis only when experimental mice passed at least 2 seconds before the orientation with a $\omega > 0$. Rearing and non-aggressive interaction (nose-to-nose contact, nose-to-body contact, passive contact and following behavior) were manually scored (experimenter blind to the viral injection). Head/body turn, stop the interaction, rearing, grooming and retreat episodes observed immediately after (the time period of 1 s) a single burst of optogenetic stimulation were manually recognized by two behavioral experts. After social behavior tests, the animals were sacrificed, and the viral infection was verified. The viral infection and the tip of the optic fiber must be localized in the region of interest (ROI) without spreading into near brain regions (PAG or Substantia Nigra, respectively). Mice that showed no precise infection or implantation during post-hoc validation were excluded from the study.

**Fiber photometry system and analyses of data.** The fiber photometry system (Doric lenses inc.) consisted of two excitation channels. A 465 nm LED (CLED_465, Doric lenses inc.) was used to extract a $Ca^{2+}$-dependent signal and a 405 nm LED (CLED_405, Doric lenses inc.) was used to obtain a $Ca^{2+}$-independent isosbestic signal. Light from the LEDs was directed through a fluorescence MiniCube composed of 4 ports with 1 integrated photodetector head (iFM-C4_AE(40 5)_E(460-490)_F(500-550) _S, Doric lenses inc.). Light emissions from GCamp6s expressing neurons were then collected back through the optic fiber (Ø

200 μm), and directed through a detection path, passing a dichroic mirror to reach the photodetector integrated in the MiniCube. A fiber photometry console (FPC, Doric lenses inc.) and the Doric software (version 5.4.1.5) was used to control the LEDs and acquire fluorescence data at 12 kHz. LEDs were alternately turned on and off at 40 Hz in a square pulse pattern.

Fiber photometry data were analyzed using custom MatLab codes (See Code Availability section). Raw signals were recorded and adjusted according to the overall trend to take account of the photo-bleaching. For each experiment we defined the $F_0$ as the 5 min baseline activity before the presentation of the social stimulus. The fluorescence change was determined as $\Delta F/F$ and calculated as $(F - F_0)/F_0$ where F is the fluorescence at each bin. The acquisition frequency was at 12 kHz (bins of 1/12,158 sec) for the entire recordings. The construction of Peri-event time histogram (PETH) was made by aligning and centering specific events. These events were obtained by manual scoring of specific behaviors at specific times to link $Ca^{2+}$ activity with the events. We used z-score to normalize $\Delta F/F$ computed as $(mean_{(\Delta F/F)} - \mu_{(\Delta F/F)})/\sigma_{(\Delta F/F)}$, where $mean_{(\Delta F/F)}$ is the averaged $\Delta F/F$ at each bin of the PETH, $\mu_{(\Delta F/F)}$ the averaged $\Delta F/F$ of the whole session and $\sigma_{(\Delta F/F)}$ the $\Delta F/F$ standard deviation of the session. A convolution using a Kernel-Gaussian sliding window of 2000 bins was then applied on the data to smooth the PETH (*gausswin* MatLab function).

**Open field**. Mice injected with rAAV5-hSyn-eYFP or rAAV5-hSyn-hChR2(H134R)-eYFP in SC and implanted with optic fiber above the VTA performed the open-field test (OF). The mice were placed in the OF arena for 10 min. The apparatus consisted in a 40 cm sided Plexiglas squared arena. The mice were randomly assigned to the optical light stimulation protocols (as previously described, Imetronic system, Pessac, France). After 10 min of free exploration the mice were placed back into their homecage. Three hours later the animals were re-tested in the OF experiment and the mice that received the light-ON epoch did not receive it and vice versa. In the end, all the animals performed both conditions. The OF test was video-tracked (Ethovision, Noldus, Wageningen, the Netherlands) to automatically obtain the distance moved and the time passed in the center. Head/body turn, rearing, grooming and retreat episodes observed immediately after (the time period of 1 s) a single burst of optogenetic stimulation were manually scored. The apparatus was cleaned using 70% ethanol after each session. After the test, the animals were sacrificed and the viral infection was verified. The viral infection and the tip of the optic fiber must be localized in the region of interest (ROI) without spreading into near brain regions (PAG or Substantia Nigra, respectively). Mice that showed no precise infection or implantation during post-hoc validation were excluded from the study.

**Real-time place preference test**. WT mice injected with rAAV5-hSyn-eYFP (control), rAAV5-hSyn-hChR2(H134R)-eYFP or rAAV5-hSyn-Jaws-KGC-GFP-ER2 in SC and implanted with optic fiber in the VTA performed the real-time place preference (rtPP). The rtPP experiment was conducted in an apparatus (spatial place preference; BioSEB) consisting of two adjacent chambers (20 × 20 × 25 cm) with dot (black) or stripe (gray) wall patterns, connected by a lateral corridor (7 × 20 × 25 cm) with transparent walls and floor. The dot chamber was always associated with a rough floor, while the stripe chamber with a smooth floor. The illumination level was uniform between the two chambers and set at 10–13 lux. Imetronic (Pessac, France) tracking software was used to track animal's movements, the time spent within each chamber and to deliver optogenetic protocols. The mice were placed in the apparatus and were free to explore both chambers for 10 min. For ChR2-expressing mice: one chamber was systematically associated with a high bursting optogenetic stimulation protocol (burst of 5 pulses of 4 msec at 20 Hz every 250 msec) while the other was not associated with any optical stimulation protocol. For Jaws-expressing mice: one chamber was associated with a linear and constant optogenetic inhibition. The chamber associated with the light stimulation/inhibition was counterbalanced to avoid any internal bias due to the cues of the compartment. The test was video-tracked (Ethovision, Noldus, Wageningen, the Netherlands) to automatically obtain the time passed in each chamber of the mice. After the test, the animals were sacrificed and the viral infection was verified. The viral infection and the tip of the optic fiber must be localized in the ROI without spreading into near brain regions (PAG or Substantia Nigra respectively). Mice that showed no precise infection or implantation during post-hoc validation were excluded from the study.

**Ex vivo slice physiology**. 200–250 μM thick horizontal midbrain slices were prepared from C57Bl/6 J WT, GAD65-cre or DAT-Cre mice. Brains were sliced by using a cutting solution containing: 90.89 mM choline chloride, 24.98 mM glucose, 25 mM $NaHCO_3$, 6.98 mM $MgCl_2$, 11.85 mM ascorbic acid, 3.09 mM sodium pyruvate, 2.49 mM KCl, 1.25 mM $NaH_2PO_4$, and 0.50 mM $CaCl_2$. Brain slices were incubated in cutting solution for 20–30 min at 35°. Subsequently, slices were transferred in artificial cerebrospinal fluid (aCSF) containing: 119 mM NaCl, 2.5 mM KCl, 1.3 mM $MgCl_2$, 2.5 mM $CaCl_2$, 1.0 mM $NaH_2PO_4$, 26.2 mM $NaHCO_3$, and 11 mM glucose, bubbled with 95% $O_2$ and 5% $CO_2$) at room temperature. Whole-cell voltage-clamp or current-clamp electrophysiological recordings were conducted at 32°–34° in aCSF (2–3 ml.min⁻¹, submerged slices). Patch pipettes were filled with a $Cs^+$-based low $Cl^-$ internal solution containing 135 mM

CsMeSO$_3$, 10 mM HEPES, 1 mM EGTA, 3.3 mM QX-314, 4 mM Mg-ATP, 0.3 mM Na$_2$-GTP, 8 mM Na$_2$-Phosphocreatine (pH 7.3 adjusted with CsOH; 295 mOsm). For ChR2-expressing WT mice, SC neurons were patched and the light stimulation protocol was assessed. For DAT-Cre and GAD65-Cre mice DAT$^+$ or GAD$^+$ neurons, respectively, of the VTA were identified as mCherry$^+$ cells. Brief pulses of blue light (10 msec) were delivered at the recording site at 10 sec intervals under the control of the acquisition software. Cells were held at −60 mV in order to evoke excitatory postsynaptic currents (EPSCs) or at 0 mV to evoke inhibitory postsynaptic currents (IPSCs). The same protocol was in DAT-Cre mice injected with the AAVrg-pCAG-FLEX-TdTomato on the NAc or DLS, and the DAT$^+$ were identified as TdTomato$^+$ cells. The delay time is calculated as the latency from the light pulse to the onset of the response. To record exclusive monosynaptic currents evoked in postsynaptic cells recorded in the whole cell at −60 mV, we combined optogenetic activation of presynaptic terminals expressing ChR2 with bath application of sodium and potassium channels blockers (1 μM tetrodotoxin, TTX, and 100 μM 4-aminopyridin, 4AP, respectively).

**Immunohistochemistry and cell counting**. Infected mice were anesthetized with pentobarbital (Streuli Pharma) and sacrificed by intracardial perfusion of 0.9% saline followed by 4% paraformaldehyde (PFA; Biochemica). Brains were post-fixed overnight in 4% PFA at 4 °C. Twenty-four hours later, they were washed with PBS and then 50 μm thick-sliced with a vibratome (Leica VT1200S).

Previously prepared slices were washed three times with phosphate-buffered saline (PBS) 0.1 M. Brain slices were pre-incubated with PBS-BSA-TX buffer (10% BSA, 0.3% Triton X-100, 0.1% NaN$_3$) for 60 min at room temperature in the dark. Subsequently, cells were incubated with primary antibodies diluted in PBS-BSA-TX (3% BSA, 0.3% Triton X-100, 0.1% NaN$_3$) overnight at 4 °C in the dark. The following day cells were washed three times with PBS 0.1 M and incubated for 60 min at room temperature in the dark with the secondary antibodies diluted in PBS-Tween buffer (0.25% Tween-20). Finally, slices were mounted using Fluoroshield mounting medium with DAPI (abcam, Cat#ab104139). In this study, the following primary antibodies were used: mouse monoclonal anti-CaMKII alpha (1/100 dilution, ThermoFisher, Cat#MA1-048, RRID: AB_325403), rabbit polyclonal anti-mCherry (1/200 dilution, abcam, Cat#ab167453) and rabbit polyclonal anti-Tyrosine Hydroxylase (1/500 dilution, abcam, Cat#ab6211). The following secondary antibodies were used at 1/500 dilution: donkey anti-mouse Alexa Fluor 488 (ThermoFisher, Cat#R37114, RRID: AB_2556542), donkey anti-rabbit Alexa Fluor 555 (ThermoFisher, Cat#A32794, RRID: AB_2762834) and donkey anti-rabbit Alexa Fluor 647 (ThermoFisher, Cat# A32795, RRID: AB_2762835). Immunostained slices were imaged using the confocal laser scanning microscopes Zeiss LSM700 and LSM800. Larger scale images were taken with the widefield Axioscan.Z1 scanner.

Cell counting of tdTomato$^+$ cells was performed on 50 μm thick SC slices from 3 ssAAV-retro/2-hSyn1-chl-tdTomato-WPRE-SV40p(A) injected mice (5 slices for each animal). The retrograde experiment was repeated with the second batch of animals injected with a CAV-cre in the VTA and an AAV-DIO-mCherry in the SC. The results of these two experiments were comparable. For each slice, images from the SC and PAG were acquired bilaterally along the whole SC dorso-ventral axis. The tdTomato$^+$ or mCherry$^+$ cells were counted automatically using MetaMorph Microscopy Software (Molecular Devices). The total percentage of cells located in the different layers of the SC or the PAG was calculated by averaging the total number of tdTomato$^+$ and mCherry$^+$ of each mouse. After CaMKIIa immunostaining onto SC slices, the number of tdTomato$^+$ or mCherry$^+$/CaMKIIa$^+$ and tdTomato$^+$ or mCherry$^+$/CaMKIIa$^-$ cells were counted.

**Viruses**. rAAV5-hSyn-eYFP (Titer ≥ 7 × 10$^{12}$ vg/mL$^{-1}$, Addgene), ssAAV-retro/2-hSyn1-chl-tdTomato-WPRE-SV40p(A) (v272-retro, Titer = 4.6 × 10$^{12}$ vg/mL$^{-1}$, Viral vector ETH Zurich), rAAV5-hSyn-hChR2(H134R)-eYFP (Titer ≥ 7 × 10$^{12}$ vg/mL$^{-1}$, Addgene), rAAV5-hSyn-Jaws-KGC-GFP-ER2 (Titer ≥ 3.8 × 10$^{12}$ vg/mL$^{-1}$, UNC Vector Core), rAAV5-Ef1α-DIO-hChR2(H134R)-eYFP (Titer ≥ 4.2 × 10$^{12}$ vg/mL$^{-1}$, UNC Vector Core), rAAV5-Ef1α-DIO-eYFP (Titer ≥ 4.2 × 10$^{12}$ vg/mL$^{-1}$, UNC Vector Core), rAAV5-hSyn-DIO-mCherry (Titer ≥ 7 × 10$^{12}$ vg/mL$^{-1}$, Addgene), AAVrg-pCAG-FLEX-tdTomato-WPRE (Titer ≥ 1 × 10$^{13}$ vg/mL$^{-1}$, Addgene), CAV-2 Cre (Titer ≥ 2.5 × 10$^{11}$ pp, Plateforme de Vectorologie de Montpellier, PVM), AAV9-hSyn-FLEX-GCamp6s-WPRE-SV40 (Titer ≥ 3.8 × 10$^{12}$ vg/mL$^{-1}$, UNC Vector Core), AAVrg-Ef1α-mCherry-IRES-Cre (Titer ≥ 7 × 10$^{12}$ vg/mL$^{-1}$, Addgene). Cholera Toxin Subunit-B (Recombinant), Alexa Fluor™ 488 Conjugate (ThermoFisher Scientific, C22841), Cholera Toxin Subunit-B (Recombinant), Alexa Fluor™ 555 Conjugate (ThermoFisher Scientific, C34776).

**Statistical analysis**. No statistical methods were used to predetermine the number of animals and cells, but suitable sample sizes were estimated based on previous experience and are similar to those generally employed in the field[19,37]. The animals were randomly assigned to each group at the moment of viral infections or behavioral tests. Statistical analysis was conducted with MatLab (The Mathwork, MATLAB_R2019b), RStudio or GraphPad Prism 7 (San Diego, CA, USA). In order to identify statistical outliers, we used the criterion Mean$_{Value}$ ± 3 × Std$_{Value}$, but no outliers were found in this study. The normality of sample distributions was assessed with the Shapiro–Wilk criterion and when violated non-parametric tests

were used. When normally distributed, the data were analyzed with independent $t$-test, paired $t$-test, while for multiple comparisons one-way ANOVA and repeated measures (RM) ANOVA were used. When normality was violated, the data were analyzed with Mann–Whitney test. For the analysis of variance with two factors (two-way ANOVA, RM two-way ANOVA and RM two-way ANOVA by both factors), normality of sample distribution was assumed, and followed by Bonferroni-Holm correction test or Bonferroni post-hoc test. All the statistical tests adopted were two-sided. When comparing two samples' distributions similarity of variances was assumed, therefore no corrections were adopted. Data are represented as the Mean ± s.e.m. and the significance was set at $P < 0.05$.

**Reporting summary**. Further information on research design is available in the Nature Research Reporting Summary linked to this article.

## Data availability
All the data are in the manuscript or in supplementary material. Source data are provided with this paper or are available in the following link: https://doi.org/10.5281/zenodo.5844936. Further data supporting the findings are available upon request. Source data are provided with this paper.

## Code availability
Innovative codes used in this study are available in the following link: https://doi.org/10.5281/zenodo.5844936.

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

## Acknowledgements

This work was supported by funds to C.B. from the Swiss National Science Foundation, NCCR SYNAPSY of the Swiss National Science Foundation, Fondation Von Meissner, Pierre Mercier, Foundation HUG and ERC consolidator grant. We are grateful to Lorena Jourdain and Mattia Lucchini for the technical support. We thank Manuel Mameli and Benoît Girard for the constructive comment on the manuscript and the entire Bellone lab for discussion.

## Author contributions

C.B., C.S., and A.Con. designed the study. C.S. and A.Con. performed the optogenetic experiments, virus injections, and optic fiber implantation. A.Con. performed the immunohistochemistry and the fiber photometry experiments and analyzed the data with the help of C.S., C.H., and A.Car. P.E., S.B., and S.M. performed electrophysiological recordings. C.B. wrote the manuscript with the help of C.S. and A.Con. A.Con prepared the figures.

## Competing interests

The authors declare no competing interests.
