## [Peer Review File · Nature Communications]

Superior Colliculus to VTA pathway controls orienting response and influences social interaction in miceREVIEWER COMMENTS

Reviewer #1 (Remarks to the Author):

This manuscript by Prevost-Solie and colleagues investigates the role of inputs from the superior colliculus to VTA dopaminergic neurons in orienting towards a social stimulus. They use viral tracing, fiber photometry, and optogenetics to show that activity within a SC-VTA-dorsolateral striatum circuit predicts and drives an orienting response towards both social and non-social stimuli. This occurs in contrast to the mPFC-VTA-NAc circuit, which drives social interaction, but not orienting. The work presented in the manuscript is technically sound, and experiments are well-controlled. This manuscript makes important advances in dissecting the circuit specificity of social behaviors, and will be of interest to neuroscientists studying social and reward behaviors.

1. The authors have demonstrated that they can drive orienting behavior by optogenetically stimulating the SC-VTA pathway, however, this does not show that this pathway is required for this behavior. Using opto- or chemo- genetics to inhibit this pathway during the orienting task and/or free social interaction and showing that orienting is impaired under these conditions would greatly strengthen the manuscript.

2. The SC pathway exhibits a greater degree of connectivity with VTA GABA neurons than VTA dopamine neurons. The SC to VTAGABA connection may be particularly important here, especially given recent results showing a role for VTA GABA neurons in controlling head angle (Hughes et al, Current Biology 2019). While I don't think this needs to be experimentally addressed, it should be discussed and the Hughes paper should be cited.

3. The studies were done in only male mice. The authors should justify use of only one sex, and should discuss how mechanisms may be the same or different in female animals.

4. (Minor) The authors should take care to not ascribe causality to fiber photometry experiments throughout the manuscript. For example, line 217 "SC-VTA pathway controls orienting responses during free social interaction" is the heading for a section that describes only photometry data, this data indicates that this pathway is activated during orienting but not that it controls orienting.

5. (Minor) Please specify the age of mice used.

Reviewer #2 (Remarks to the Author):

The authors show that projections arising from the mouse superior colliculus targeting the ventral tegmental area are engaged in reorientation during social interaction.

Nicely written but some concerns emerge.

Major issues

1. My main concern is that SC-VTA activity may not be causal to the control of head orientation to the appearance of a conspecific. The SC will respond to a novel visual stimulus appearing in the periphery whether it is a conspecific or a prey etc. I suspect that the SC might simply attempt to 'inform' VTA neurons of anticipated movement planning just like it does to its neighbouring SNc through axonal collaterals.

To assess SC-VTA's role in head control during social interaction, one would have to demonstrate loss of function in behavior during circuit perturbation. A standard way to do this is to silence the SC-VTA connection (eg with Jaws, NpHR etc) during the orientation task and evaluate differences at first appearance of the conspecific.

The reason this is crucial is due to the fact that processes shown in Fig 1b-b' may be fibres en passage travelling to downstream target structures heading to the oculomotor centres, caudal pons or medulla giving off collaterals to VTA. Are there any axons labelled with eYFP in the caudal pons (with ref to Fig 1b)? This would suggest an efferent copy is sent to VTA and place its role as a spectator of action rather than its cause. We have seen this in the past in the case of SNc, in that crossed projections arising from the intermediate layer of SC targeting specifically the contralateral medulla give off collaterals onto SNc neurons. This would suggest an efferent copy is sent to VTA and place its role as a spectator of action rather than its cause.

The authors demonstrate a functional connection however it may be that the SC is attempting to inform the VTA (as in the case of SNc) about what is about to happen rather than controlling the movement itself. One way to ensure a connection is by identifying synaptic contact by expressing GFP selectively in synaptic structures (eg SynGFP) and show colocalization onto TH+/VTA+ somata or close appositions.

Furthermore, if this is the case, are there any axons labelled with mCherry in the caudal pons?

2. Quantification of animal behavior during orientation task is more qualitative than quantitative.

I recommend that kinematic profiles or plotting instantaneous head orientation $\omega(t)$ (as defined in fig 2c') be presented on top of all calcium traces (i.e. Fig 2e-g, Fig 4, etc). It is difficult to assess stimulus novelty and correlation to calcium activity in the course of the entire encounter with the conspecific during recordings without this information. I would suggest placing a video recording (in supplement) of the typical interaction between subject and conspecific to help the reader to better understand the dynamics underlying the interaction.

3. During injections into VTA, how do the authors ensure that viral spread has not entered into SNc, a dopaminergic area that is known to also receive SC input. Injection sites should be properly documented and the criteria for exclusion clearly specified. For instance, can we see the CAV-Cre injection zone in VTA in Fig. 1c? It is critical to ensure locality in these injections.

4. Determining mediolaterality of SC activity is important to help understand the resultant head movement, be it horizontal or vertical. I see in Fig 1d that neurons are labelled mainly medially and the optic fibre placement is located also medially (Fig 2b). This area is known to drive vertical head movements and not horizontal, yet a horizontal movement is required in order to orient to the conspecific which is depicted throughout the study in terms of ω . Redgrave and other groups have shown the segregation between medial to lateral in the rodent SC; In fact the authors also cite Redgrave's work (Comoli et al;ref 14) on line 87. Please elaborate.

I would also like to ask if any retrogradely infected neurons are found in the lateral regions of the intermediate layer of SC, which comprise the crossed tectoreticular pathway and are in fact involved in generated horizontal gaze/head orienting movement, as would be the case in these experiments.

Just to clarify: The authors use the abbreviation eSC in Fig1d. Is this meant to indicate the Superficial Layer? please define abbrev in legend. What does the coloured region in orange indicate in Fig 1d (right)?

5. When applying the perturbation (Fig 5), there is a lack of specificity in the expression of ChR2 in the SC meaning that optical stimulation in VTA can be unspecific. Using a dual vector strategy as in Fig 1c to target specific SC-VTA would overcome this limitation, ergo ambiguity.

What is the diameter of the optic fiber implant that the authors used? I could not find it. I suspect 200um (from the lesion in Fig 3b) but cannot be sure. Obviously if larger diameters are used, the undesired effect of non-specific stimulation is more likely. I am concerning because fibers of passage will get activated during stimulation and the interpretation of the perturbation will not be clear. We boil down to my first main concern.

6. Other issues

a. Please insert actual current recordings in Fig 1i and Fig 7g. It is odd to not include them in the main figure and place them in the supplement.

b. I also urge the authors to use bath application of TTX & 4AP to determine whether evoked currents are monosynaptic rather than using latency between stim onset and current onset.

c. I would use, but do not insist, transynaptic rabies to dissect the contribution of the SC to the diverging VTA-DLS & VTA-NAc pathways and reveal their identity. This comment is with regard to Fig 7.

Minor

1. Please revise lines 263-265. Confusing
2. Line 271. Use plural 'mice'
3. Line 30, spell correlates correctly
4. Fig 1g. please define the scale bars for recordings. I can't find any info in the legends.
5. When citing that SC controls spatial attention (line 87), please include correct refs (eg work from Krauzlis).

Reviewer #3 (Remarks to the Author):

This paper examines the role of the superior colliculus projection the VTA in social orienting responses. They show evidence of the SC-VTA projection mediating orienting responses to social stimuli via fiber photometry and optogenetic stimulation. They also suggest that the downstream cells mediating this response are DLS-projecting DAT cells in the VTA. Overall, these data were interesting but the claims that the SC-VTA-DLS projection is specifically involved in social orienting are not sufficiently supported. In addition, throughout the paper the claims should be edited to agree more closely with the data.

Major concerns:

Fig 1 - The authors claim that their responses are monosynaptic based solely on latency. First, there are no details in the methods about how the latency is calculated - is it latency from light pulse to onset of the response, or latency from light pulse to peak of the response? These details should be included in methods. Additionally, to make the claim that these responses are monosynaptic, the authors should repeat the experiments in TTX/4AP. Otherwise, these claims should be removed. Furthermore, please include some example traces of the responding cells so that readers can see what the responses look like.

Figure 2- The ipsi- vs contra- distinction in selectivity is not directly tested with a statistical test despite the claims in the text. It should be tested directly for both the social target and the ball.

Fig 3 - There are no details in the methods about how the behaviors were classified (rearing, head/body turn, etc). Please add these details including any validation that was performed on this classification. Also, it would strengthen the paper to include a similar inanimate object control in this experiment to solidify claims that the SC-VTA pathway is involved in social orienting specifically.

Fig 4 - The authors make claims that the SC-VTA pathway is involved in social orienting in freely moving behavior, but the data shown involves social contact, not orienting. If the tracking data from this experiment exists, it should be possible to extract social orienting times and align the photometry data to these times.

Fig 7 - The authors show in Figure 1 that GAD+ cells receive more frequent input than DAT+ cells, but then choose to only record from DAT+ cells in Figure 7. Why not record from GAD+ cells here, as well? They also use retrograde AAV to attempt to label DAT+ neurons in VTA, even though retrograde AAV does not typically work well to label DAT+ neurons. Potentially, low uptake of the retroAAV virus by NAc-projecting DAT neurons could be playing a role in the fact that the authors saw few responses in NAc-projecting DAT neurons. It would be helpful to see some example images of the histology to ensure that the expression of this retroAAV in DAT projection neurons. Again the authors use latency to claim that the responses are monosynaptic. These claims should either be changed from "directly connected" cells to "affected" cells, or they should include an experiment in TTX/4AP showing that these connections are definitely monosynaptic.

Fig 8 - It isn't clear why the authors switched their data presentation of the laser effects in e and f compared to earlier figures. Please plot the data showing effects in minute 1 and minute 2, as was shown in Figures 3 and 5. It also would make sense to see some orienting data from this experiment, rather than looking only at social interaction time, given the emphasis of the paper overall on social orienting. Finally, it is confusing that the authors conclude these 2 dopamine projections control different aspects of social behavior. They instead seem to be showing that both projections regulate social interaction time, but modulate it in different directions.

-In line 273 and other places, the authors refer to "non-social exploratory behavior". I think they may mean "locomotion" here, which should be made explicit. In other words, they need to define what they mean. On a similar note, what is the difference between Figure 6c and 6d? Is this the same data analyzed with two different statistical methods? This should be clarified or one of these

should be removed.

-It was helpful to see the effects on ball orienting were in fact similar to social orienting in an early figure, but unclear why the object comparison was absent in later figures. It should be included and implying that these results are specific to social orienting should be omitted (unless the data suggests otherwise).

Minor concerns:

-There is a typo in line 172 (latest should be lasted).

-No statistical tests are reported for Figure 7, though the authors make claims that DLS-projecting DAT neurons are receiving the primary input from SC, and not NAc-projecting cells. Please add some statistical test to support this claim.

-Fig 5 - In panel e, some animals appear to have fewer instances of stimulation, can the authors please explain why?

-Please include labels on the x-axis in Figure 2 (e-g, right), rather than only in the figure legend. Also please include x-axis labels on Figure 4 in d,f,h,i,j,k.

Reviewer #4 (Remarks to the Author):

Prevoste-Solie et al characterize the role of the ventral tegmental area (VTA) in modulating super colliculus (SC) during orientation to social and salient stimuli in male mice. A combination of viral tracing, electrophysiology, photometric, and optogenetic approaches are used. The authors show that the SC projects to specific populations of VTA dopamine and GABA neurons regulate orientation towards conspecific mice, and that manipulation of these projections disrupt orienting behavior to other conspecifics. They next show that SC projections are uniquely coupled to VTA-dorsolateral striatum (DLS) projections. Furthermore, patterns of activity during social interaction are not similarly displayed by mPFC to VTA projections, which appear preferentially involved in engagement and maintenance of social episodes rather than orientation to social stimuli. The authors conclude that SC neurons projecting to the VTA are uniquely involved in the orientation of animals to salient stimuli, and such projections target a DLS projecting subpopulation of neurons. Overall, this is a comprehensive and impactful study. Study design is sound and interpretations are supported by the data. Authors are commended for clear presentation of data and in particular, presentation of individual heatmaps in some figures, which are quite informative. There are, however, several methodological and interpretive concerns that should be addressed:

1. Social activity/behaviour is measured during the light cycle when animals wouldn't normally be active or awake. This is a concern in the context of the present study because there is a large literature indicating that activity of SC neurons change during mouse wakefulness (e.g. Triplett, 2018; Zhang et al, 2019). Authors do not explain why they did not perform the study during naturally active cycle when rodents are likely to engage in social behavior. This concern and potential implications should be discussed.

2. There appears to be baseline differences between the control and Chr2 groups in Figure 2F. Statistics are performed within groups but if statistics were performed across groups would the Chr2 group 'off' or 'on' period be significantly higher than eYFP? This concern is not trivial because data in this figure are fundamental to some of the main conclusions of the manuscript.

3. Key technical details regarding the fiber photometry system are not included. Was an isosbestic 405 nm (or comparable) channel utilized to control for motion artifact? What filters were used? Reporting light power during experiments and numerical aperture would also be informative for readers. How was the bleach detrending performed? Linear fit? Polynomial? Would comparable results be obtained with a different z-score window?

4. The authors utilize the AAVretrograde serotype for DA projections. Has this approach been validated for retrograde DA neuron tracing? This is a concern because there are relatively few neurons in the tracing studies.

5. mCherry appears to be expressing outside the VTA which is the tag utilized for the retrograde

Cre transport. Does this raise concern for SC and mPFC recordings in Figure 2 not being purely from projections to VTA but also more ventral regions?

6. The lack of overall PFC response in Fig 2 may be from increased diversity of response in the global signal. Were recordings exclusively from IL or PL subregions of the mPFC or both? This is important given the documented divergence of these subregions in animal behavior. A hitmap of fiber placements may help with this. Similarly, in Fig 8 the NAC subjects have a lower social interaction time in the 'off' period compared to the DLS subjects. This difference in the off period may drive the differences in the index scores used for cross group comparison. Thus it is possible the light stimulation doesn't really produce opposing effects based on projection. It may be informative to do a 3x2 (projection by light) ANOVA to clarify if this effect is driven by light off vs light on period.

7. In Fig 5 it appears that some results during free exploration are not the same as those obtained in the social interaction "task" in Fig 3. A significant effect of time (habituation) is observed for the ChR mice, which was not seen in the task in Fig 3. Thus it may be that the more rich engagement allowed by the free exploration procedures may attenuate some of the deficits in SC-VTA stim seen in the social task.

8. There seems to be extra-VTA expression into the nigra in Fig 7 raising the possibility that ChR2+ neurons outside the VTA are stimulated. A hitmap of fibers for the optogenetic studies should be shown (same in Fig 8).

9. How were cohorts/replicates counterbalanced? Where mPFC animals from the same cohort as SC animals or were cohorts more homogenous?

10. Place preference appears to be done for only one day. It is possible that aversive effects would have emerged in a retention test. This is a concern given that some animals start to show a place avoidance in supplemental Fig 4, and that there is a main effect of light and virus.

11. Only male mice were used. This should be mentioned in the abstract and discussion.

Minor

The social orientation task isn't really a task. There is nothing the animals need to do or learn/solve. It would be more accurate to just call it a test or assay.

Methods suggest signals in photometry experiments were somal but this can't be certain unless mini-scopes or a soma targeting GCaMP were utilized.

Please report number of excluded data points from figures based on the outlier criteria.

There should be justification for why the Bonferroni correction is used sometimes and the more liberal Bonferroni-holm correction is used other times.

Scalebar in Figure 2B-1 is larger than 2B-2 but the legend says they are the same size and magnification.

Do NAC and DLS projections have different effects on opto-stimulation caused behavior like those seen in Figures 2 and

SC to VTA projections are thought to be primarily excitatory. However, stimulating this pathway disrupts social orientation. This finding is somewhat counterintuitive – would authors expect that inhibiting this pathways will improve or have no effect on orientation? A brief discussion of this phenomenon would be helpful.

REVIEWER COMMENTS

Reviewer #1 (Remarks to the Author):

This manuscript by Prevost-Solie and colleagues investigates the role of inputs from the superior colliculus to VTA dopaminergic neurons in orienting towards a social stimulus. They use viral tracing, fiber photometry, and optogenetics to show that activity within a SC-VTA-dorsolateral striatum circuit predicts and drives an orienting response towards both social and non-social stimuli. This occurs in contrast to the mPFC-VTA-NAc circuit, which drives social interaction, but not orienting. The work presented in the manuscript is technically sound, and experiments are well-controlled. This manuscript makes important advances in dissecting the circuit specificity of social behaviors, and will be of interest to neuroscientists studying social and reward behaviors.

1. The authors have demonstrated that they can drive orienting behavior by optogenetically stimulating the SC-VTA pathway, however, this does not show that this pathway is required for this behavior. Using opto- or chemo- genetics to inhibit this pathway during the orienting task and/or free social interaction and showing that orienting is impaired under these conditions would greatly strengthen the manuscript.

We thank the reviewer for the comments and suggestions. We now added new optogenetic inhibition experiments using Jaws in Figure 3 and Supp. Figure 4 and 5. We observed that inhibition of SC to VTA pathway disrupted habituation in the time passed in the frontal field during orientation test and increased the time spent interacting with the conspecific in the free social interaction test. It is important to mention that in this case the protocol of inhibition was applied continuously. For this reason, it has not been possible to analyse the behavioural responses elicited by a burst of light as we did for the Chr2 condition. This point is also discussed in the discussion

2. The SC pathway exhibits a greater degree of connectivity with VTA GABA neurons than VTA dopamine neurons. The SC to VTA GABA connection may be particularly important here, especially given recent results showing a role for VTA GABA neurons in controlling head angle (Hughes et al, Current Biology 2019). While I don't think this needs to be experimentally addressed, it should be discussed and the Hughes paper should be cited.

We agree with the comments of the reviewer. In the new version of the manuscript we discussed the SC - VTA GABA neurons connectivity (page 17, lines 6-19) and added the suggested reference.

3. The studies were done in only male mice. The authors should justify use of only one sex, and should discuss how mechanisms may be the same or different in female animals.

In our study, we decided to focus on one component of the social behaviour, the social orientation, through dissecting the SC - VTA pathway and more particularly VTA DA neurons. It is known now that the oestrus cycle can modify the VTA DA activity (Calipari et al., Nat Comm 2017) and can influence social behaviour. Thereby we decided to use only male. However, we agree with the reviewer that it should be justified in the manuscript and describe how the mechanism could be similar or not. We modify the manuscript in the discussion according to this comment (page 16, lines 20-21).

4. (Minor) The authors should take care to not ascribe causality to fiber photometry experiments throughout the manuscript. For example, line 217 “SC-VTA pathway controls orienting responses during free social interaction” is the heading for a section that describes only photometry data, this data indicates that this pathway is activated during orienting but not that it controls orienting.

We thank the reviewer for the comment. We adapt the manuscript accordingly.

5. (Minor) Please specify the age of mice used.

We specified this information in the materials and methods chapter (page 19, line 3).

Reviewer #2 (Remarks to the Author):

The authors show that projections arising from the mouse superior colliculus targeting the ventral tegmental area are engaged in reorientation during social interaction.

Nicely written but some concerns emerge.

Major issues

1. My main concern is that SC-VTA activity may not be causal to the control of head orientation to the appearance of a conspecific. The SC will respond to a novel visual stimulus appearing in the periphery whether it is a conspecific or a prey etc. I suspect that the SC might simply attempt to 'inform' VTA neurons of anticipated movement planning just like it does to its neighbouring SNc through axonal collaterals.

To assess SC-VTA's role in head control during social interaction, one would have to demonstrate loss of function in behavior during circuit perturbation. A standard way to do this is to silence the SC-VTA connection (eg with Jaws, NpHR etc) during the orientation task and evaluate differences at first appearance of the conspecific.

We thank the reviewer for these suggestions. We now added optogenetic inhibition experiments using Jaws in Figure 3 and Supp. Figure 4 and 5. We observed that inhibition of SC to VTA pathway disrupted habituation in the time passed in the frontal field during orientation test and increased the time spent interacting with the conspecific in the free social interaction test. It is important to mention that in this case the protocol of inhibition was applied continuously. For this reason, it has not been possible to analyse the behavioural responses elicited by a burst of light as we did for the ChR2 condition.

The reason this is crucial is due to the fact that processes shown in Fig 1b-b' may be fibres en passage travelling to downstream target structures heading to the oculomotor centres, caudal pons or medulla giving off collaterals to VTA. Are there any axons labelled with eYFP in the caudal pons (with ref to Fig 1b)? This would suggest an efferent copy is sent to VTA and place its role as a spectator of action rather than its cause. We have seen this in the past in the case of SNc, in that crossed projections arising from the intermediate layer of SC targeting specifically the contralateral medulla give off collaterals onto SNc neurons. This would suggest an efferent copy is sent to VTA and place its role as a spectator of action rather than its cause.

We agree with the reviewer about this important point. We checked closely the caudal pons and no fibers were found in this structure from our side. However, the Allen Brain Atlas (<https://connectivity.brain-map.org>) clearly showed that after a unilateral injection of anterograde virus in the SC (similar coordinates used in our study) labelled fibers were detected in the VTA region and in deeper structures (see pictures below). We estimate that

the point raised by the reviewer is of interest and we added a paragraph in the manuscript to discuss this possibility (page 16-17, lines 23-4).

The authors demonstrate a functional connection however it may be that the SC is attempting to inform the VTA (as in the case of SNc) about what is about to happen rather than controlling the movement itself. One way to ensure a connection is by identifying synaptic contact by expressing GFP selectively in synaptic structures (eg SynGFP) and show colocalization onto TH+/VTA+ somata or close appositions. Furthermore, if this is the case, are there any axons labelled with mCherry in the caudal pons?

To prove functional connectivity between SC and VTA DA neurons we performed new experiments and used patch-clamp to show that the VTA DA cells receive direct projection from the SC. In this revised version, we decided to identify monosynaptic inputs by using 4-AP and TTX while patching VTA DA neurons. As the reviewer can see in figure 1i, VTA DA neurons receive monosynaptic connections from SC. We hope this new set of experiments will satisfy and convince the reviewer about the functional connectivity between SC and VTA.

2. Quantification of animal behavior during orientation task is more qualitative than quantitative.

I recommend that kinematic profiles or plotting instantaneous head orientation $\omega(t)$ (as defined in fig 2c') be presented on top of all calcium traces (i.e. Fig 2e-g, Fig 4, etc). It is difficult to assess stimulus novelty and correlation to calcium activity in the course of the entire encounter with the conspecific during recordings without this information. I would suggest placing a video recording (in supplement) of the typical interaction between subject and conspecific to help the reader to better understand the dynamics underlying the interaction.

We appreciate the suggestions and we agree that this information could help the reader to better understand our results. We added calcium traces and head orientation in figure 2, and calcium traces, head orientation and nose-to-body distance in figure 4. As the reviewer can

see, there is evidence of the correlation between the head orientation and the calcium activity.

3. During injections into VTA, how do the authors ensure that viral spread has not entered into SNc, a dopaminergic area that is known to also receive SC input. Injection sites should be properly documented and the criteria for exclusion clearly specified. For instance, can we see the CAV-Cre injection zone in VTA in Fig. 1c? It is critical to ensure locality in these injections.

We agree with the reviewer that checking the infection zone when using retrograde labelling technique is a critical point. In this revised version of the manuscript, we repeated the experiment injecting AAV-retrograde-TdTomato in the VTA and checking the TdTomato expression in the SC. We paid attention to eventual infected cells in the SNc, and we took into consideration for the analysis only brains that did not show spreading in regions near the VTA (see exclusion criteria). The results revealed similar results observed previously in the CAV-cre batch of animals. We added details in the material and methods.

4. Determining mediolaterality of SC activity is important to help understand the resultant head movement, be it horizontal or vertical. I see in Fig 1d that neurons are labelled mainly medially and the optic fibre placement is located also medially (Fig 2b). This area is known to drive vertical head movements and not horizontal, yet a horizontal movement is required in order to orient to the conspecific which is depicted throughout the study in terms of ω . Redgrave and other groups have shown the segregation between medial to lateral in the rodent SC; In fact the authors also cite Redgrave's work (Comoli et al;ref 14) on line 87. Please elaborate.

We agree that a medio-lateral segregation exists in the SC as well as a dorso-ventral segregation between deep layers and superficial layers. For fiber photometry experiments, we decided to implant the fiber optic in medial part of the SC since in figure 1, we observed a higher number of VTA projecting cells using retrograde viral strategy. In our paradigm, animals are free to move and to use both vertical and horizontal movements. Since we did not observe an increased activity of the SC-VTA projecting neurons during rearing behaviour, we assumed that this pathway is not active during vertical movement. On the other hand, head horizontal movements and body turn elicits a response in the SC-VTA pathway.

I would also like to ask if any retrogradely infected neurons are found in the lateral regions of the intermediate layer of SC, which comprise the crossed tectoreticular pathway and are in fact involved in generated horizontal gaze/head orienting movement, as would be the case in these experiments.

The localization of the SC-VTA projecting neurons in the SC is represented in figure 1d. The majority of the cells are in the medial part and only few are located in the lateral SC.

Just to clarify: The authors use the abbreviation eSC in Fig1d. Is this meant to indicate the Superficial Layer? please define abbrev in legend. What does the coloured region in orange indicate in Fig 1d (right)?

We agree with the reviewers that our abbreviations and coloured regions were not explicit enough. We changed that in the new version of the manuscript and we hope that it is clearer for the reviewer.

5. When applying the perturbation (Fig 5), there is a lack of specificity in the expression of Chr2 in the SC meaning that optical stimulation in VTA can be unspecific. Using a dual vector strategy as in Fig 1c to target specific SC-VTA would overcome this limitation, ergo ambiguity.

We thank the reviewer for the comment and to get the opportunity to clarify our viral strategy. We first thought to use the strategy described by the reviewer, however we decided to inject Chr2 in the SC and stimulate the terminals to avoid any possibility to stimulate neurons in the SC sending collaterals in other structures. We therefore thought this viral strategy was more appropriate to avoid any unwanted effects and loss of specificity in the SC-VTA pathway. We verified post-hoc the site of infection within the SC and we have excluded all the mice which infection was outside the structure.

What is the diameter of the optic fiber implant that the authors used? I could not find it. I suspect 200um (from the lesion in Fig 3b) but cannot be sure. Obviously if larger diameters are used, the undesired effect of non-specific stimulation is more likely. I am concerning because fibers of passage will get activated during stimulation and the interpretation of the perturbation will not be clear. We boil down to my first main concern.

We are sorry this information was not present in the previous version of the manuscript. Indeed, we used 200um fiber to prevent undesired effects of a wide stimulation of the zone. We thank the reviewer for this comment and the diameter of the fiber is now specified in the current version.

6. Other issues

- a. Please insert actual current recordings in Fig 1i and Fig 7g. It is odd to not include them in the main figure and place them in the supplement.

We now included current recordings in the main figures instead of supplementary.

- b. I also urge the authors to use bath application of TTX & 4AP to determine whether evoked currents are monosynaptic rather than using latency between stim onset and current onset.

We agree with the reviewer and as written in a previous response, we now used TTX and 4-AP and showed that evoked currents are monosynaptic.

- c. I would use, but do not insist, transynaptic rabies to dissect the contribution of the SC to the diverging VTA-DLS & VTA-NAc pathways and reveal their identity. This comment is with regard to Fig 7.

We thank the reviewer for this suggestion and agreed that using a trans-synaptic strategy with rabies virus would be interesting to dissect the two pathways. Unfortunately, we did not have the authorization to use such a technique within our laboratory and were not able to use it for our study.

Minor

1. Please revise lines 263-265. Confusing
We modified it in the current version of the manuscript.
2. Line 271. Use plural 'mice'
We changed in the current version of the manuscript.
3. Line 30, spell correlates correctly
We changed in the current version of the manuscript.
4. Fig 1g. please define the scale bars for recordings. I can't find any info in the legends.
We added details on the scale bar in the figure directly.
5. When citing that SC controls spatial attention (line 87), please include correct refs (eg work from Krauzlis).
We thank the reviewer for this comment and we are sorry for this mistake. We now include good references.

Reviewer #3 (Remarks to the Author):

This paper examines the role of the superior colliculus projection the VTA in social orienting responses. They show evidence of the SC-VTA projection mediating orienting responses to social stimuli via fiber photometry and optogenetic stimulation. They also suggest that the downstream cells mediating this response are DLS-projecting DAT cells in the VTA. Overall, these data were interesting but the claims that the SC-VTA-DLS projection is specifically involved in social orienting are not sufficiently supported. In addition, throughout the paper the claims should be edited to agree more closely with the data.

Major concerns:

Fig 1 - The authors claim that their responses are monosynaptic based solely on latency. First, there are no details in the methods about how the latency is calculated - is it latency from light pulse to onset of the response, or latency from light pulse to peak of the response? These details should be included in methods.

We thank the reviewer for the constructive comment and we are sorry that it was not clear how latency was calculated. The delay time is calculated as the latency from light pulse to the onset of the response. We now included these details in material methods and hope it is clearer for the reviewer (page 27, lines 25-26).

Additionally, to make the claim that these responses are monosynaptic, the authors should repeat the experiments in TTX/4AP. Otherwise, these claims should be removed. Furthermore, please include some example traces of the responding cells so that readers can see what the responses look like.

We agree that looking at the delay only to confirm monosynaptic inputs was not enough to prove functional connectivity between SC and VTA DA neurons. In the revised version, we decided to identify monosynaptic inputs by using 4-AP and TTX while patching VTA DA neurons. As the reviewer can see in figure 1j, VTA DA neurons receive monosynaptic connections from SC. We hope this new set of experiments will satisfy and convince the reviewer about the functional connectivity between SC and VTA.

Figure 2- The ipsi- vs contra- distinction in selectivity is not directly tested with a statistical test despite the claims in the text. It should be tested directly for both the social target and the ball.

In the text, we did not indicate that the calcium signals recorded in the ipsi- are different from the ones measured in the contra-orientation (page 7, lines 25-28 of the new version of the manuscript). The text read: *No significant increase in normalized $\Delta F/F$ was observed*

during contra-recorded orienting response or passive crossing events. The statistical test used (RM One-way ANOVA followed by Bonferroni Holm correction) support our claims.

Fig 3 - There are no details in the methods about how the behaviors were classified (rearing, head/body turn, etc). Please add these details including any validation that was performed on this classification. Also, it would strengthen the paper to include a similar inanimate object control in this experiment to solidify claims that the SC-VTA pathway is involved in social orienting specifically.

We are sorry our classification was not clear to the reviewer. We re-wrote this part of the text in material and method session (pages 23-24, lines 26-17) and we hope it is clearer in the current version of the manuscript. We performed a new opto-orientation test in a new batch of mice using a moving object as stimulus. Similarly to our previous findings obtained in the social paradigm, we observed that ChR2-expressing mice tend to pass less time oriented towards the moving ball during the first minute of the light ON epoch (Sup. Figure 2). Our data therefore suggest that the SC-VTA pathway is involved in orienting behaviour toward salient moving stimuli.

Fig 4 - The authors make claims that the SC-VTA pathway is involved in social orienting in freely moving behavior, but the data shown involves social contact, not orienting. If the tracking data from this experiment exists, it should be possible to extract social orienting times and align the photometry data to these times.

We thank the reviewer for the useful suggestion. We now added the analysis of head orientation during the free social interaction test and aligned with the calcium activity and the distance between the nose of the experimental animal and the gravity center of the conspecific (Figure 4c). As the reviewer can appreciate, we still observed an increase in the activity of the SC-VTA pathway in proximal and distal ipsi-orientations during free social interaction. These data are consistent with the data observed in the orientation test.

Fig 7 - The authors show in Figure 1 that GAD+ cells receive more frequent input than DAT+ cells, but then choose to only record from DAT+ cells in Figure 7. Why not record from GAD+ cells here, as well? They also use retrograde AAV to attempt to label DAT+ neurons in VTA, even though retrograde AAV does not typically work well to label DAT+ neurons. Potentially, low uptake of the retroAAV virus by NAc-projecting DAT neurons could be playing a role in the fact that the authors saw few responses in NAc-projecting DAT neurons. It would be helpful to see some example images of the histology to ensure that the expression of this retroAAV in DAT projection neurons. Again the authors use latency to claim that the responses are monosynaptic. These claims should either be changed from “directly connected” cells to “affected” cells, or they should include an experiment in TTX/4AP showing that these connections are definitely monosynaptic.

In the figure 7, we focused exclusively on DAT+ cells. A previous study (Zhou, Z. et al. 2019) has shown that VTA GABA cells receiving SC inputs were preferentially projecting to the amygdala. This pathway is highly involved in flight and escape behaviours. Furthermore, with the social aspect of this study we therefore decided to focus on VTA DA neurons, known to be involved in social behaviours.

To be sure of the expression of the AAVrg virus and the tropism of DAT+ neurons, we repeated injection of AAVrg in the DLS and in the new version of the manuscript we present images of the colocalization of cre-positive TH neurons in the VTA that projects to the DLS (figure 7 e-e'). To prove the monosynaptic connectivity between SC and VTA DA neurons that project to the DLS we performed patch clamp using 4-AP and TTX (figure 7f). We hope this new set of experiments will satisfy and convince the reviewer.

Fig 8 - It isn't clear why the authors switched their data presentation of the laser effects in e and f compared to earlier figures. Please plot the data showing effects in minute 1 and minute 2, as was shown in Figures 3 and 5. It also would make sense to see some orienting data from this experiment, rather than looking only at social interaction time, given the emphasis of the paper overall on social orienting. Finally, it is confusing that the authors conclude these 2 dopamine projections control different aspects of social behavior. They instead seem to be showing that both projections regulate social interaction time, but modulate it in different directions.

We thank the reviewer for the nice suggestions. We now homogenized the data presentations and plotted minute 1 and 2 for all the experiments and we hope it is clearer in the current version. Finally, we gave our interpretation of these results in a new paragraph in the discussion chapter (page 17, lines 6-19).

- In line 273 and other places, the authors refer to “non-social exploratory behavior”. I think they may mean “locomotion” here, which should be made explicit. In other words, they need to define what they mean. On a similar note, what is the difference between Figure 6c and 6d? Is this the same data analyzed with two different statistical methods? This should be clarified or one of these should be removed.

We thank the reviewer for her/his comments and in the current version of the manuscript we now wrote “locomotion”. As the reviewer noted, old figures 6c and 6d present the same data but in different manner. We decided to present only a graph in order to avoid repetition and clarify the main points.

- It was helpful to see the effects on ball orienting were in fact similar to social orienting in an early figure, but unclear why the object comparison was absent in later figures. It should be included and implying that these results are specific to social orienting should be omitted (unless the data suggests otherwise).

We thank the reviewer for giving us the opportunity to clarify the message of our study. Indeed, we did not state that these results were specific to social orienting and we are sorry if it was not clear in the text. We now adapt the manuscript accordingly to avoid any misinterpretation of our data. Importantly, it is not possible for us to include data about the interaction with a moving ball, because unfortunately the animals show aversion to this stimulus after some minutes. Since the valence of the moving ball is different from the conspecific, it is plausible to imagine that different mechanisms would play a role during interaction.

Minor concerns:

- There is a typo in line 172 (latest should be lasted).
We made the change in the current version.
- No statistical tests are reported for Figure 7, though the authors make claims that DLS-projecting DAT neurons are receiving the primary input from SC, and not NAc-projecting cells. Please add some statistical test to support this claim.
The N for the SC-VTA-NAc group is equal to 1, as a consequence, no statistical test could be performed. Find more cells for this group is extremely rare and we think that the presented graphs are supporting our claims.
- Fig 5 - In panel e, some animals appear to have fewer instances of stimulation, can the authors please explain why?
The reviewer is right. All the animals passed 2 minutes in the arena with the conspecific and the optostimulation, but not all the stimulations were recognisable on the videos. The analysis and quantification of the induced behaviours were conducted post-hoc during the manual scoring of the videos. If the optostimulation was not clearly visible, it was not possible to recognize the induced behaviour, for this reason we present less episodes for some animals.
- Please include labels on the x-axis in Figure 2 (e-g, right), rather than only in the figure legend. Also please include x-axis labels on Figure 4 in d,f,h,i,j,k.
We now added the labels on the x-axis for these panels.

Reviewer #4 (Remarks to the Author):

Prevoste-Solie et al characterize the role of the ventral tegmental area (VTA) in modulating super colliculus (SC) during orientation to social and salient stimuli in male mice. A combination of viral tracing, electrophysiology, photometric, and optogenetic approaches are used. The authors show that the SC projects to specific populations of VTA dopamine and GABA neurons regulate orientation towards conspecific mice, and that manipulation of these projections disrupt orienting behavior to other conspecifics. They next show that SC projections are uniquely coupled to VTA-dorsolateral striatum (DLS) projections. Furthermore, patterns of activity during social interaction are not similarly displayed by mPFC to VTA projections, which appear preferentially involved in engagement and maintenance of social episodes rather than orientation to social stimuli. The authors conclude that SC neurons projecting to the VTA are uniquely involved in the orientation of animals to salient stimuli, and such projections target a DLS projecting subpopulation of neurons. Overall, this is a comprehensive and impactful study. Study design is sound and interpretations are supported by the data. Authors are commended for clear presentation of data and in particular, presentation of individual heatmaps in some figures, which are quite informative. There are, however, several methodological and interpretive concerns that should be addressed:

We thank the reviewer for supportive and nice comments on our study.

1. Social activity/behaviour is measured during the light cycle when animals wouldn't normally be active or awake. This is a concern in the context of the present study because there is a large literature indicating that activity of SC neurons change during mouse wakefulness (e.g. Triplett, 2018; Zhang et al, 2019). Authors do not explain why they did not perform the study during naturally active cycle when rodents are likely to engage in social behavior. This concern and potential implications should be discussed.

We agree with the reviewer that it should have been discussed in the text.

In our institute, for technical reasons, we have no possibility for the moment to perform study during a naturally active cycle. This concern has been now discussed in the manuscript (pages 17, line 13-21).

2. There appears to be baseline differences between the control and ChR2 groups in Figure 2F. Statistics are performed within groups but if statistics were performed across groups would the ChR2 group 'off' or 'on' period be significantly higher than eYFP? This concern is not trivial because data in this figure are fundamental to some of the main conclusions of the manuscript.

We agree that baselines between on and off groups are not identical, however when performing statistical tests, no significant differences emerged between eYFP and ChR2/Jaws baselines (OFF condition).

3. Key technical details regarding the fiber photometry system are not included. Was an isosbestic 405 nm (or comparable) channel utilized to control for motion artifact? What filters were used? Reporting light power during experiments and numerical aperture would also be informative for readers. How was the bleach detrending performed? Linear fit? Polynomial? Would comparable results be obtained with a different z-score window?

We thank the reviewer for the suggestions. We now added all technical aspects of the fiber photometry protocols in the Material and Methods chapter (pages 24-25, lines 19-2).

4. The authors utilize the AAVretrograde serotype for DA projections. Has this approach been validated for retrograde DA neuron tracing? This is a concern because there are relatively few neurons in the tracing studies.

To be sure of the expression of the AAVrg virus and the tropism of DAT+ neurons, we repeated injection of AAVrg in the DLS and in the new version of the manuscript we present images of the colocalization of cre-positive TH neurons in the VTA that projects to the DLS (figure 7 e-e'). Regarding the quantity of neurons, we agree that they are not a big amount, but we demonstrated that they are DAT-positive and connected with the SC. We hope that this validation will satisfy and convince the reviewer.

5. mCherry appears to be expressing outside the VTA which is the tag utilized for the retrograde Cre transport. Does this raise concern for SC and mPFC recordings in Figure 2 not being purely from projections to VTA but also more ventral regions?

In our knowledge and according to the Allen Brain Atlas (<https://connectivity.brain-map.org>, see picture below) the mPFC strongly projects to the VTA. It is possible to see some projections in ventral regions but these are not comparable with the fibers observed in the VTA.

6. The lack of overall PFC response in Fig 2 may be from increased diversity of response in the global signal. Were recordings exclusively from IL or PL subregions of the mPFC or both? This is important given the documented divergence of these subregions in animal behavior. A hitmap of fiber placements may help with this. Similarly, in Fig 8 the NAC subjects have a lower social interaction time in the ‘off’ period compared to the DLS subjects. This difference in the off period may drive the differences in the index scores used for cross group comparison. Thus it is possible the light stimulation doesn’t really produce opposing effects based on projection. It may be informative to do a 3x2 (projection by light) ANOVA to clarify if this effect is driven by light off vs light on period.

We thank the reviewer for his/her comment. We injected the virus and implanted the optic fibers in the prelimbic cortex (PrL) and all the site of injections were checked post-hoc. However, since the infralimbic cortex (IL) was also labelled and is above the tip of the optic fiber, we do not exclude that the recordings could be influenced by the activity of both brain regions. For this region, we prefer to report in the manuscript that we record the mPFC. Furthermore, it is important to note that, all animals performed both tests: the free social interaction and orientation test and in both tests, they showed clear and consistent calcium signals (this information is now been included in of the manuscript page 20, line 17-18).

We agree that baselines are not identical. However, when performing statistical tests, no significant differences emerged between eYFP and ChR2 baselines (3x2 ANOVA. eYFP^{OFF} vs NAc^{OFF}: $p = 0.5114$. eYFP^{OFF} vs DLS^{OFF}: $p = 0.1142$).

7. In Fig 5 it appears that some results during free exploration are not the same as those obtained in the social interaction “task” in Fig 3. A significant effect of time (habituation) is observed for the ChR mice, which was not seen in the task in Fig 3. Thus it may be that the more rich engagement allowed by the free exploration procedures may attenuate some of the deficits in SC-VTA stim seen in the social task.

We agree with the reviewer to this point and we would like to strength the fact that the two tests are very different and therefore we cannot make direct comparison. As

the reviewer suggested, during the direct interaction task there is a richer engagement of the stimulus to the environment and this could explain the difference in the habituation.

8. There seems to be extra-VTA expression into the nigra in Fig 7 raising the possibility that ChR2+ neurons outside the VTA are stimulated. A hitmap of fibers for the optogenetic studies should be shown.

We thank the reviewer for the nice suggestion. We now added a heatmap of the fibers placement in the VTA (Sup. Fig. 2a).

9. How were cohorts/replicates counterbalanced? Where mPFC animals from the same cohort as SC animals or were cohorts more homogenous?

mPFC and SC mice were assigned randomly from the same cohorts. Surgeries and experiments were performed at the same time for both groups to avoid any biases. We have now specified this in the manuscript (page 8, lines 6-9).

10. Place preference appears to be done for only one day. It is possible that aversive effects would have emerged in a retention test. This is a concern given that some animals start to show a place avoidance in supplemental Fig 4, and that there is a main effect of light and virus.

We now added place preference with mice injected with Jaws. In that case, there is no main effect of light and virus. In the ChR2 group only main effect of the virus was present. It could be possible that aversion could emerge by repeating the test through several days. However, all our tests (free social interaction and orientation) were done acutely, preventing these effects to arise.

11. Only male mice were used. This should be mentioned in the abstract and discussion.

We now discuss the use of only male in the manuscript (page 16, lines 20-21) and add this information in the abstract (page 2, lines 5-6).

Minor

The social orientation task isn't really a task. There is nothing the animals need to do or learn/solve. It would be more accurate to just call it a test or assay.

We thank the reviewer for the suggestion. We now call this test "social orientation test" in the manuscript.

Methods suggest signals in photometry experiments were somal but this can't be certain unless mini-scopes or a soma targeting GCaMP were utilized.

We agree with this comment. We removed the term somal to better clarify what we are seeing.

Please report the number of excluded data points from figures based on the outlier criteria.

Based on the outlier criteria, we did not exclude any point from this study. We report this information in the current version of the manuscript (page 29-30, lines 29-1).

There should be justification for why the Bonferroni correction is used sometimes and the more liberal Bonferroni-holm correction is used other times.

To be sure of the effect of some optogenetic stimulation or inhibition experiments, we decided to apply the Bonferroni correction to avoid any false positive effect. Indeed, since the manipulation of the neuronal circuitry using optogenetic and the manual scoring of the behaviour can bring some biases, this prevented us to misinterpret some data.

Scalebar in Figure 2B-1 is larger than 2B-2 but the legend says they are the same size and magnification.

We corrected the figures accordingly.

Do NAC and DLS projections have different effects on opto-stimulation caused behavior like those seen in Figures 2 and SC to VTA projections are thought to be primarily excitatory. However, stimulating this pathway disrupts social orientation. This finding is somewhat counterintuitive – would authors expect that inhibiting this pathways will improve or have no effect on orientation? A brief discussion of this phenomenon would be helpful.

We now added experiments using Jaws in the SC-VTA pathway. As the reviewer can see, data show that inhibiting this pathway induce loss of habituation in the orientation test and higher social interaction in the free social interaction.

REVIEWER COMMENTS

Reviewer #1 (Remarks to the Author):

The authors have satisfied my concerns, only one minor concern remains.

1. Please check that all electrophysiology traces are correctly labelled-in figure 1i, one would expect that a monosynaptic connection would result in no observable current in the "ptx+ttx" condition but a restoration of the current in the "ptx+ttx+4-AP" condition. As currently labelled, it appears the opposite is occurring.

Reviewer #2 (Remarks to the Author):

The authors have addressed my concerns point-to-point in a satisfactory way.

Some remarks.

Fig 1g (left trace) appears to be a recording performed in current clamp. If this is the case, the units should be in mV and not pA. Please change accordingly.

It appears that the authors confused the color-coded correspondence of the traces during TTX+4AP experiments. In both SC-VTA-DAT+ (blue) and SC-VTA-GAD+ (red) cases in Fig 1i, the flat signal should correspond to the PTX+TTX and not the PTX+TTX+4AP; and vice versa. If this is not a typo, which I doubt, then I am confused about what the authors claim to be monosynaptic. Maybe I am missing it, but please state the stats regarding the recordings, e.g. amplitude in pA's.

Overall, there has been significant improvement to the original manuscript.

Reviewer #3 (Remarks to the Author):

I think the paper is improved, especially with the confirmation of monosynaptic connectivity and increasing consistency of plotting/analysis across figures. The paper is impressive in terms of how much ground is covered (ie how many experiments). However, I still find some of the main points of the paper unclear and I suggest the authors consider rewriting/reorganizing the paper to clarify so the takeaway message is clear and accurate. Here are some of the things that remain unclear to me that could be addressed (without experiments):

-If the argument is not that this SC-VTA projection is involved specifically in social orienting behavior, and instead it is involved in orienting more generally, why does the title, abstract, and introduction emphasize social orienting?

-If the VTA DA projection to DLS is such an essential downstream effector of the observed effects of SC to VTA stimulation, why is the effect on social interaction time the opposite, and why weren't the same orienting readouts plotted the same way for this projection as well?

-If the loss of function experiment is so critical that it was suggested by some of the reviewers and results are described in the abstract, why is it only a supp figure?

-More generally, what do the authors consider the relationship between social orienting and social interaction time? I think I don't really understand this and therefore find this sentence in abstract (and similar points throughout paper) confusing: "stimulation of SC-VTA pathway promotes head/body movements associated with decreasing social interaction, constant inhibition of this pathway alters the habituation of head orientation towards conspecific and increases social interaction."

Reviewer #4 (Remarks to the Author):

The authors have thoughtfully addressed all my major concerns. This is now a comprehensive and well-controlled dataset that should have a substantial impact on the field.

REVIEWER COMMENTS

Reviewer #1 (Remarks to the Author):

The authors have satisfied my concerns, only one minor concern remains.

1. Please check that all electrophysiology traces are correctly labelled-in figure 1i, one would expect that a monosynaptic connection would result in no observable current in the "ptx+ttx" condition but a restoration of the current in the "ptx+ttx+4-AP" condition. As currently labelled, it appears the opposite is occurring.

We thank the reviewer for this comment. Indeed, it is a mistake in the labelling. We would like to apologize and in the current version, the mislabelling has been corrected.

Reviewer #2 (Remarks to the Author):

The authors have addressed my concerns point-to-point in a satisfactory way.

Some remarks.

Fig 1g (left trace) appears to be a recording performed in current clamp. If this is the case, the units should be in mV and not pA. Please change accordingly.

We thank the reviewer to notice this mistake. We now replaced pA in mV on the figure 1g.

It appears that the authors confused the color-coded correspondence of the traces during TTX+4AP experiments. In both SC-VTA-DAT+ (blue) and SC-VTA-GAD+ (red) cases in Fig 1i, the flat signal should correspond to the PTX+TTX and not the PTX+TTX+4AP; and vice versa.

If this is not a typo, which I doubt, then I am confused about what the authors claim to be monosynaptic. Maybe I am missing it, but please state the stats regarding the recordings, e.g. amplitude in pA's.

Indeed, we mislabelled the traces and we inverted the traces between PTX+TTX+4AP and PTX+TTX. We now corrected in the new version of the figure 1.

Overall, there has been significant improvement to the original manuscript.

We would like to thank the reviewer for all the comments that helped us to improve the manuscript.

Reviewer #3 (Remarks to the Author):

I think the paper is improved, especially with the confirmation of monosynaptic connectivity and increasing consistency of plotting/analysis across figures. The paper is impressive in terms of how much ground is covered (ie how many experiments). However, I still find some of the main points of the paper unclear and I suggest the authors consider rewriting/reorganizing the paper to clarify so the takeaway message is clear and accurate. Here are some of the things that remain unclear to me that could be addressed (without experiments):

We thank the reviewer for the nice comments about our manuscript.

-If the argument is not that this SC-VTA projection is involved specifically in social orienting behavior, and instead it is involved in orienting more generally, why does the title, abstract, and introduction emphasize social orienting?

In our manuscript, we demonstrate that the SC to VTA pathway is implicated in orienting response toward salient stimuli and we focus on social behaviour because it is a highly salient experience. To clarify the message of the paper, we decided to change the title. The title now reads: "Superior Colliculus to VTA pathway controls orienting response and influences social interaction".

Although social orientation is an important element in social motivation hypothesis underlying Autism Spectrum Disorders, the importance of social orienting responses in social interaction has been largely ignored. These elements have been discussed in the discussion session. We now added a new sentence in the correspondent paragraph to underlying this point (page 15, lines 21-25).

-If the VTA DA projection to DLS is such an essential downstream effector of the observed effects of SC to VTA stimulation, why is the effect on social interaction time the opposite, and why weren't the same orienting readouts plotted the same way for this projection as well?

By optogenetically stimulating VTA-DA to DLS pathway, we show a decrease in social interaction, that is consistent with the observations of the SC-VTA pathway stimulation. However, the VTA-DA to NAc stimulation increases the time interacting with the conspecific, as presented in figure 7I.

The orienting response to social stimulus during stimulation of VTA-DA to DLS and VTA-DA to NAc pathways is presented in Supp Figure 6. Contrary to the experiments presented in figure 5, where we stimulated SC to VTA pathway, here we did not observe changes in the time passed in the frontal field between OFF and ON conditions. These data suggest that DA neurons projecting to DLS are therefore *per se* not involved in head rotation and orienting response. In the discussion we reported that it has been shown that GABA neurons in the VTA would encode head angles for each of the three principal axes of rotation, and it has been suggested that the control of head rotation is essential for most motivated behaviours. The apparent inconsistency of the data could be explained therefore by the fact that when we stimulate SC to VTA pathway, we activate both DA and GABA neurons. Altogether, our findings suggest that SC to VTA triggers orienting behaviour that are essential for downstream regulated motivated behaviours. This part is now better explained in the discussion session (page 17, 18 to 22)

If the loss of function experiment is so critical that it was suggested by some of the reviewers and results are described in the abstract, why is it only a supp figure?

We decided to show in the main figure the orientation test data for the jaws experiment (figure 3 f) while to put the rest of the data in supplementary matherial to simplify the message of the manuscript and to avoid too crowded figures. If the editor and the reviewer consider we should put these data in the main figures, we will be happy to modify the figures accordingly.

-More generally, what do the authors consider the relationship between social orienting and social interaction time? I think I don't really understand this and therefore find this sentence in abstract (and similar points throughout paper) confusing: "stimulation of SC-VTA pathway promotes head/body movements associated with decreasing social interaction, constant inhibition of this pathway alters the habituation of head orientation towards conspecific and increases social interaction."

Stimulation of SC to VTA pathway increases head body turns and decreases social interaction. Furthermore, we observed that ChR2-expressing mice stopped the action that they were performing. These data suggest that altering the pattern of activity of SC to VTA pathway by imposing a stimulation protocol, alters the orienting response toward the salient stimulus and that this alteration results in a decrease of social interaction. The data obtained in the direct interaction test suggest that the SC to VTA pathway is recruited during head/body turn and controls orientation. This orienting response is at the basis of social interaction. In support of this hypothesis, photostimulation induced less following and nose-to-nose interaction (Sup. Figures 4h-j), while photoinhibition elicited opposite changes in following behaviour between light ON and OFF conditions (Sup. Figures 4k-m). We have now simplified the sentence in the abstract (Page 2 lines 11 to 14). Furthermore, we added additional explanations in the result session (page 10 lines 24 to 28).

Reviewer #4 (Remarks to the Author):

The authors have thoughtfully addressed all my major concerns. This is now a comprehensive and well-controlled dataset that should have a substantial impact on the field.

We thank the reviewer for the nice comments on our manuscript.